# BevSplat: Resolving Height Ambiguity via Feature-Based Gaussian Primitives for Weakly-Supervised Cross-View Localization

Qiwei Wang[1], Shaoxun Wu[1], Yujiao Shi[1]

[1]ShanghaiTech University, Shanghai, China. Correspondence to: Yujiao Shi
<shiyj2@shanghaitech.edu.cn>

## Abstract

This paper addresses the problem of weakly supervised cross-view localization, where the goal is to estimate the pose of a ground camera relative to a satellite image with noisy ground truth annotations. A common approach to bridge the cross-view domain gap for pose estimation is Bird's-Eye View (BEV) synthesis. However, existing methods struggle with height ambiguity due to the lack of depth information in ground images and satellite height maps. Because a single 2D pixel could represent points at various depths and heights, its true 3D position is ambiguous. Previous solutions either assume a flat ground plane or rely on complex models, such as cross-view transformers. We propose BevSplat, a novel method that resolves height ambiguity by using feature-based Gaussian primitives. Each pixel in the ground image is represented by a 3D Gaussian with semantic and spatial features, which are synthesized into a BEV feature map for relative pose estimation. We validate our method on the widely used KITTI and VIGOR datasets, which include both pinhole and panoramic query images. Experimental results show that BevSplat significantly improves localization accuracy over prior approaches. Our code is available at https://github.com/wangqww/BevSplat.

## 1 Inotroduction

Cross-view localization, the task of estimating the pose of a ground camera with respect to a satellite or aerial image, is a critical problem in computer vision and remote sensing. This task is especially important for applications such as autonomous driving, urban planning, and geospatial analysis, where accurately aligning ground-level and satellite views is crucial. However, it presents significant challenges due to the inherent differences in scale, perspective, and environmental context between ground-level images and satellite views.

To navigate these complexities, particularly the common difficulty of acquiring precise ground-truth (GT) camera locations at scale, weakly supervised learning [1, 2] has recently emerged as a promising paradigm. In this setting, models are trained using only noisy annotations, such as approximate camera locations with errors potentially reaching tens of meters, which adds another layer of complexity to the task. Nevertheless, the primary advantage of weak supervision lies in its ability to leverage less labor-intensive data collection, making it a more scalable and practical approach for many real-world applications where extensive precise annotations are infeasible.

A key strategy to address cross-view localization is Bird's-Eye View (BEV) synthesis [3–5, 1, 6], which generates a bird's-eye view representation from the ground-level image. The BEV image can then be compared directly to a satellite image, facilitating relative pose estimation. However, existing methods often rely on Inverse Perspective Mapping (IPM), which assumes a flat ground plane [1, 6],

39th Conference on Neural Information Processing Systems (NeurIPS 2025).

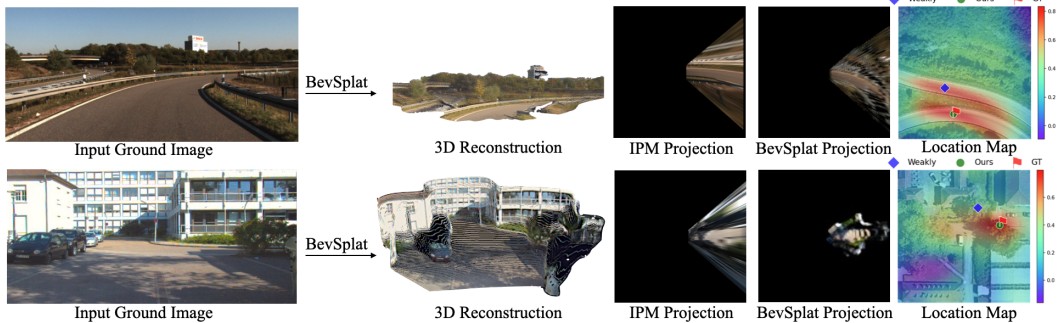

Figure 1: Our BevSplat cross-view localization process begins by using a depth prediction network on a single ground-level image to acquire its depth map. This depth map is then employed for 3D reconstruction into Gaussian splats, which are finally projected to a Bird's-Eye View (BEV). In comparison to the Inverse Perspective Mapping (IPM) approach, our method demonstrates improved recovery of BEV curves, more effective handling of building occlusions, and enhanced practical localization performance.

or on high-complexity models like cross-view transformers [3–5] to address height ambiguity, the challenge of resolving the elevation difference between the ground and satellite views.

The flat terrain assumption used in IPM leads to the loss of critical scene information above the ground plane and introduces distortions for objects farther from the camera, as shown in Fig. 1. On the other hand, while cross-view transformers are effective at handling distortions and objects above the ground plane, they are computationally expensive. Furthermore, in weakly supervised settings, noisy ground camera pose annotations provide weak supervision, making it difficult for high-complexity models like transformers to converge [1], ultimately leading to suboptimal localization performance.

In this paper, we propose BevSplat to address these challenges. BevSplat generates feature-based 3D Gaussian primitives for BEV synthesis. Unlike previous 3D Gaussian Splatting (3DGS) [7] methods that rely on color-based representations, we represent each pixel in the ground-level image as a 3D Gaussian with semantic and spatial features. These Gaussians are associated with attributes such as position in 3D space, scale, rotation, and density, which are synthesized into a BEV feature map using a visibility-aware rendering algorithm that supports anisotropic splatting. This approach enables us to handle height ambiguity and complex cross-view occlusions, improving the alignment between the ground-level image and the satellite view for more accurate pose estimation, without the need for expensive depth sensors or complex model architectures.

We validate our approach on the widely used KITTI and VIGOR datasets, where the former localizes images captured by pin-hole cameras, and the latter aims to localize panoramic images, demonstrating that the proposed BevSplat significantly outperforms existing techniques in terms of localization accuracy in various localization scenarios.

## 2 Related Work

### 2.1 Cross-view Localization

Cross-view localization, which aligns ground-level images with satellite imagery, has evolved from image retrieval to fine-grained pose estimation. Early approaches framed this as an image retrieval task, using metric learning to match ground queries to satellite image slices [8–12]. While modern transformers have improved retrieval performance, practical application remains challenging [13, 14]. Many recent methods adopt a coarse-to-fine pipeline. This typically involves a coarse retrieval step [15] followed by fine-grained (pixel-level) localization to identify the precise camera pose [16–18, 3, 19, 5, 20–22]. A key limitation of these methods is their reliance on precise, GPS-based training data, which is often prone to inaccuracies. To overcome this, weakly supervised settings have been proposed to learn from noisy pose annotations [1, 2]. These weakly supervised settings differ in their assumptions. [2] assumes the availability of GT labels in a source domain and access to cross-view pairs in the target domain. In contrast, [1] addresses a more challenging scenario where

source domain GT labels are unavailable and no target domain pairs are accessible. In this work, we tackle the same task setting as [1].

## 2.2 Bird's-Eye View Synthesis

BEV synthesis, which generates bird's-eye view images from ground-level perspectives, has been widely applied to cross-view localization. While LiDAR and Radar sensors offer high accuracy for localization tasks [23–26], their high cost limits their use. For camera-only systems, multi-camera setups are commonly employed [27–30], primarily focusing on tasks like segmentation and recognition. In localization, methods like Inverse Perspective Mapping (IMP) assume a flat ground plane for BEV synthesis [1, 6, 20, 18], which can be overly simplistic for complex environments. Transformer-based models address these challenges but struggle with weak supervision and noisy pose annotations [3–5]. While methods such as [31, 32] also employ feature Gaussians for the image-to-BEV transformation, they typically benefit from rich depth and semantic information afforded by multi-sensor setups. While effective in some contexts, they face limitations in resource-constrained, real-world scenarios. In stark contrast, our approach is constrained to rely exclusively on weakly supervised signals derived purely from images, presenting a considerably more challenging task.

## 2.3 Sparse-View 3D Reconstruction

In our method, we adopt algorithms similar to 3D reconstruction to represent ground scenes. Sparse-view 3D reconstruction has been a major focus of the community. Nerf-based approaches [33] and their adaptations [34] have shown the potential for single-view 3D reconstruction, though their application is limited by small-scale scenes and high computational cost. Recent works using diffusion models [35–37] and 3D Gaussian representations [7, 38–40], as well as transformer- and Gaussian-based models[41, 42], have achieved sparse-view 3D reconstruction on a larger scale, but the complexity of these models still restricts their use due to computational demands. Approaches like [43–45] leverage pre-trained models to directly generate Gaussian primitives, avoiding the limitations of complex models while enabling scene reconstruction from sparse views. We apply such methods to single-view reconstruction, achieving high-accuracy cross-view localization.

# 3 Method

In this paper, we address cross-view localization by aligning ground-level and satellite images under weak supervision, where initial ground camera locations are only approximate. Our objective is to accurately estimate camera pose from these noisy priors by leveraging Gaussian primitives, which effectively manage height ambiguity and enable efficient generation of Bird's-Eye View (BEV) feature maps. First, we employ an orientation prediction network analogous to that in G2SWeakly to align the orientations of the ground and satellite images. Subsequently, our BevSplat method lifts the ground image to 3D (Section 3.1) and projects the corresponding Feature Gaussians into the BEV perspective to render the ground-view BEV features (Section 3.2.1). Finally, these features are compared against the satellite features by computing a similarity score (Section 3.2.2).

## 3.1 Geometric Gaussian Primitives Generation

Inspired by 3D Gaussian Splatting (3DGS) [7], we represent the 3D scene as a collection of Gaussian primitives. Our generation process first establishes their initial geometry and appearance. Given the inherent difficulty of directly learning accurate depth in our weakly supervised framework, we utilize a pre-trained depth estimation model to predict per-pixel depth $D_i$ from the ground-level image. The initial 3D coordinate $\mu_i$ for primitives associated with each pixel $(u_i, v_i)$ is then determined from $D_i$ and the specific camera model.

For pinhole cameras, $\mu_i$ is computed by back-projecting 2D image coordinates $(u_i, v_i)$ using depth $D_i$ and camera intrinsics $K$ as $\mu_i = K^{-1} D_i [u_i, v_i, 1]^T$.

For panoramic cameras, where pixel coordinates $(u_i, v_i)$ represent viewing angles (e.g., azimuth $u_i$ and polar angle $v_i$), the initial 3D coordinate $\mu_i$ is obtained by scaling the depth $D_i$ along a unit direction vector $\hat{\mathbf{d}}_i = [x_i, y_i, z_i]^T$. The components are defined as:

$$x_i = -\sin(v_i)\cos(u_i), \quad y_i = -\cos(v_i), \quad z_i = -\sin(v_i)\sin(u_i). \tag{1}$$

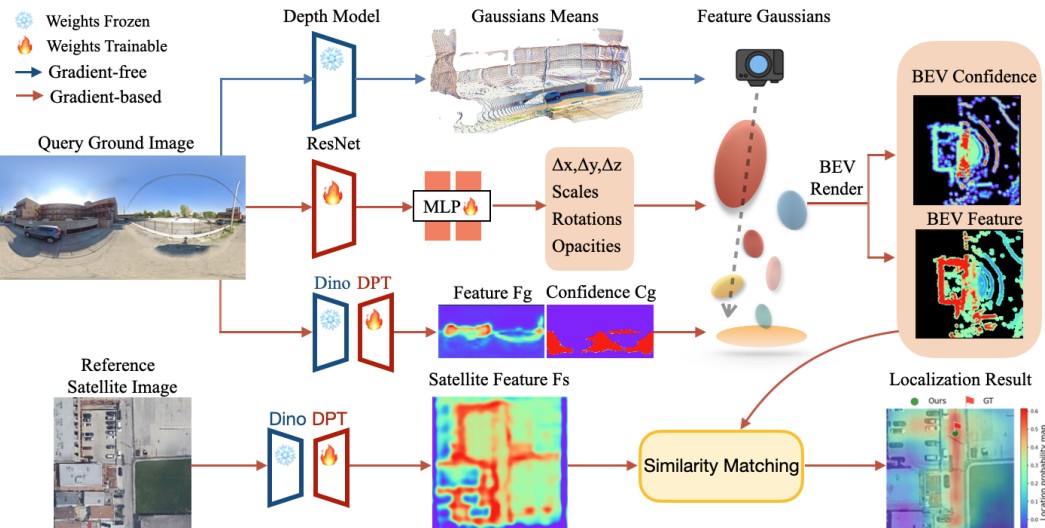

Figure 2: **BevSplat Framework Overview.** Query ground image Gaussian primitive initialization involves: (1) A pre-trained depth model for initial 3D positions ($\mu_i$). (2) A ResNet and MLP to predict offsets ($\Delta\mathbf{p}_k$), scale ($\mathbf{S}_k$), rotation ($\mathbf{R}_k$), and opacity ($O_k$). (3) A DPT-fine-tuned DINOv2 for extracting semantic features ($\mathbf{f}_i$) and confidences ($c_i$), which are then bound to these Gaussians. These feature Gaussians are subsequently rendered into BEV feature and confidence maps. Satellite image features are extracted using an identical DINOv2-DPT backbone (note: weights are shared for KITTI but differ for VIGOR, similar to G2SWeakly [1]). Localization is achieved by matching satellite features with the rendered query BEV features via cosine similarity within a sliding window.

The initial 3D coordinate is thus $\mu_i = D_i \hat{\mathbf{d}}_i$.

Subsequently, a ResNet [46] extracts local features $\mathbf{f}_{\text{loc}}^i$ for each pixel $i$ from the ground-level image. These features serve as input to a multi-layer perceptron (MLP [47]), denoted $F_{gs}$, which predicts attributes for $N_p = 3$ distinct Gaussian primitives originating from each pixel. The predicted attributes for each of these $N_p$ Gaussian primitives include positional offsets $\Delta\mathbf{p}_k = (\Delta x_k, \Delta y_k, \Delta z_k)$ relative to $\mu_i$, an anisotropic scale $\mathbf{S}_k$, a rotation quaternion $\mathbf{R}_k$, and an opacity value $O_k$. Predicting multiple primitives per pixel enhances single-image representation density. The collection of parameters $G_i$ for these $N_p$ primitives from pixel $i$, incorporating their final 3D positions, is:

$$G_i = \{(\mathbf{S}_k, \mathbf{R}_k, O_k, \mu_i + \Delta x_k, \mu_i + \Delta y_k, \mu_i + \Delta z_k)\}_{k=1}^{N_p}. \tag{2}$$

Here, $k$ indexes the $N_p$ primitives associated with that pixel, the set of parameters $\{(\mathbf{S}_k, \mathbf{R}_k, O_k, \Delta x_k, \Delta y_k, \Delta z_k)\}_{k=1}^{N_p}$ is the direct output of $F_{gs}(\mathbf{f}_{\text{loc}}^i)$ and $\mathbf{f}_{\text{loc}}^i$ is the ResNet feature vector for pixel $i$. This process yields an initial set of geometric and appearance-based Gaussian primitives. These primitives are subsequently enriched with semantic features for localization, as detailed next.

## 3.2 Feature-based Gaussian Primitives for Relative Pose Estimation

For robust semantic feature extraction from ground and satellite images, inspired by [43, 48, 44], we fine-tune a pre-trained DINOv2 [49] model augmented with a DPT [50] module. From the ground image, this pipeline yields a feature map $\mathbf{F}_g \in \mathbb{R}^{H_g \times W_g \times C}$ and a confidence map $\mathbf{C}_g \in \mathbb{R}^{H_g \times W_g \times 1}$. The confidence map $\mathbf{C}_g$, derived from $\mathbf{F}_g$ via an additional convolutional layer and a sigmoid activation, assigns lower weights to dynamic objects (e.g., vehicles) and higher weights to static elements (e.g., road surfaces), indicating feature reliability for localization. For the predominantly static satellite image, we solely extract its feature map $\mathbf{F}_s \in \mathbb{R}^{H_s \times W_s \times C}$.

### 3.2.1 BEV Feature Rendering

The extracted ground features $\mathbf{F}_g$ and confidences $\mathbf{C}_g$ are then bound to the Gaussian primitives generated as described in Section 3.1. Specifically, each of the $N_p = 3$ Gaussian primitives originating

from a ground image pixel $i$ is augmented with the corresponding per-pixel feature vector $\mathbf{f}_i$ (sampled from $\mathbf{F}_g$) and confidence score $c_i$ (from $\mathbf{C}_g$). All $N_p$ primitives derived from the same pixel $i$ thus share identical $\mathbf{f}_i$ and $c_i$ values, effectively embedding semantic information and its reliability into the 3D representation.

Next, viewing the scene from a BEV perspective (camera directed downwards), the features $\mathbf{f}_b$ and confidences $c_b$ bound to each Gaussian primitive $b$ are rendered onto a 2D plane. This differentiable $\alpha$-blending process, analogous to RGB rendering in 3DGS [7], yields the BEV feature map $\mathbf{F}_{BEV}$ and confidence map $\mathbf{C}_{BEV}$:

$$\mathbf{F}_{BEV} = \sum_{b=1}^{\mathcal{N}_{\mathcal{G}}} \mathbf{f}_b \alpha_b T_b, \quad \mathbf{C}_{BEV} = \sum_{b=1}^{\mathcal{N}_{\mathcal{G}}} c_b \alpha_b T_b, \tag{3}$$

where primitives $b \in \{1, \ldots, \mathcal{N}_{\mathcal{G}}\}$ are sorted by depth from the BEV camera. Here, $T_b = \prod_{j=1}^{b-1}(1 - \alpha_j)$, $\mathcal{N}_{\mathcal{G}}$ is the total number of ground-image Gaussian primitives, $\alpha_b$ is the opacity of primitive $b$, and $\mathbf{f}_b, c_b$ are its bound feature vector and confidence score (inherited from its source pixel), respectively.

### 3.2.2 Pose Estimation via Confidence-Guided Similarity Learning

The location probability map $\mathbf{P}(u, v)$, representing the similarity between satellite image features $\mathbf{F}_s(u, v)$ and confidence-weighted ground BEV features $\mathbf{C}_{BEV}\mathbf{F}_{BEV}$, is computed as:

$$\mathbf{P}(u, v) = \frac{\langle \mathbf{F}_s(u, v), \mathbf{C}_{BEV}\mathbf{F}_{BEV} \rangle}{\|\mathbf{F}_s(u, v)\| \cdot \|\mathbf{C}_{BEV}\mathbf{F}_{BEV}\|}. \tag{4}$$

Here, $\mathbf{F}_s$ denotes the satellite image features, while $\mathbf{F}_{BEV}$ and $\mathbf{C}_{BEV}$ are the BEV features and confidence map derived from the ground image, respectively. $\| \cdot \|$ signifies the $L_2$ norm.

**Supervision.** Following [1], a deep metric learning objective supervises the network. For each query ground image, we compute location probability maps: $\mathbf{P}_{\text{pos}}$ against its positive satellite image and $\mathbf{P}_{\text{neg,idx}}$ against each of the $M$ negative satellite images. The weakly supervised loss $\mathcal{L}_{\text{Weakly}}$ aims to maximize the peak of $\mathbf{P}_{\text{pos}}$ while minimizing the peak for each $\mathbf{P}_{\text{neg,idx}}$:

$$\mathcal{L}_{\text{Weakly}} = \frac{1}{M} \sum_{\text{idx}=1}^{M} \log \left( 1 + e^{\alpha \left[ \text{Peak}(\mathbf{P}_{\text{neg,idx}}) - \text{Peak}(\mathbf{P}_{\text{pos}}) \right]} \right), \tag{5}$$

where the hyperparameter $\alpha$ (set to 10) controls convergence speed.

If more accurate (though potentially noisy) location labels $(x^*, y^*)$ are available during training (e.g., GPS with error up to $d = 5$ meters, where $\beta$ is the ground resolution in m/pixel of $\mathbf{P}_{\text{pos}}$), an auxiliary loss $\mathcal{L}_{\text{GPS}}$ is introduced:

$$\mathcal{L}_{\text{GPS}} = \left| \text{Peak}(\mathbf{P}_{\text{pos}}) - \text{Peak}(\mathbf{P}_{\text{pos}}[x^* \pm d/\beta, y^* \pm d/\beta]) \right|. \tag{6}$$

This objective encourages the global peak of $\mathbf{P}_{\text{pos}}$ to align with the local peak probability found within the $d$-meter radius neighborhood of the noisy label $(x^*, y^*)$.

The total optimization objective is then:

$$\mathcal{L}_{all} = \mathcal{L}_{\text{Weakly}} + \lambda_1 \mathcal{L}_{\text{GPS}}, \tag{7}$$

where $\lambda_1 = 1$ if noisy location labels are utilized during training, and $\lambda_1 = 0$ otherwise.

## 4 Experiments

In this section, we first describe the benchmark datasets and evaluation metrics for evaluating the effectiveness of cross-view localization models, followed by implementation details of our method. Subsequently, we compare our method with state-of-the-art approaches and conduct experiments to demonstrate the necessity of each component of the proposed method.

**KITTI dataset**. The KITTI dataset [51] consists of ground-level images captured by a forward-facing pinhole camera with a restricted field of view, complemented by aerial images [52], where each

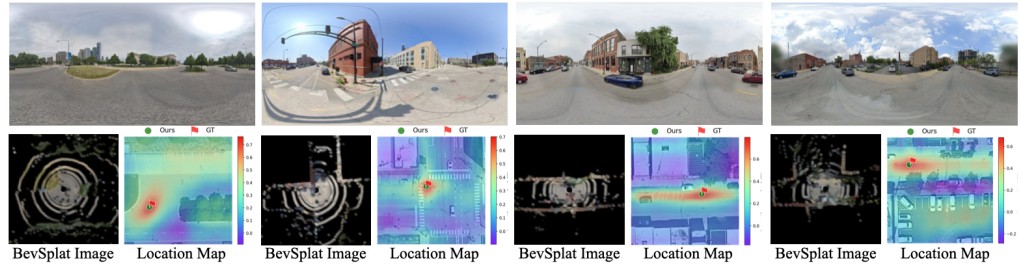

Figure 3: Visualization of the query ground image (up) and the estimated relative pose with respect to the satellite image (bottom right) on VIGOR dataset. The BEV image projected from the query ground image using the estimated Gaussian primitives is presented in the bottom left for each example.

aerial patch covers a ground area of approximately $100 \times 100\mathrm{m}^2$. The dataset includes a training set and two test sets(Same-Area and Cross-Area). For the Same-Area test set, the test query images are from the same geographical regions as the training set, but are not the same images. For the Cross-Area test set, the test images come from entirely new geographical regions that were not seen during training, testing the model's ability to generalize. The location search range of ground images is approximately $56 \times 56\mathrm{m}^2$, with an orientation noise of $\pm 10°$.

**VIGOR dataset**. The VIGOR dataset [15] includes geo-tagged ground panoramas and satellite images from four US cities: Chicago, New York, San Francisco, and Seattle. Each satellite patch spans $70 \times 70\mathrm{m}^2$ and is labeled positive if the ground camera is within its central $1/4$ region; otherwise, it is semi-positive. The dataset also has Same-Area and Cross-Area splits: Same-Area uses training and testing data from the same region, while Cross-Area splits training and testing between two separate city groups. We use only positive satellite images for all experiments, following [1].

**Evaluation Metrics**. For the KITTI dataset [51], we evaluated localization and orientation errors by calculating mean and median errors in meters and degrees, respectively. We also compute recall at thresholds of 1 m and 3 m for longitudinal (along the driving direction) and lateral (orthogonal to the driving direction) localization errors, as well as 1 ° and 3 ° for orientation errors. A localization is considered successful if the estimated position falls within the threshold of the ground truth, and an orientation is accurate if its error is within the angle threshold. For the VIGOR dataset [15], which does not provide driving direction information, we report mean and median errors as outlined in [1].

**Visualization.** We provide visualizations of the query images and localization results in Fig.3. For better clarity, we show the synthesized BEV image generated from our estimated Gaussian primitives at the bottom left of each example (though the model uses BEV feature maps for localization). Further qualitative results, including an analysis of failure cases, are provided in the supplementary material.

**Implementation Details.** For 3D point cloud generation from ground images, we employ specific depth estimation models: DepthAnythingV2 [53] for the pinhole camera images of the KITTI dataset [51], and UniK3D [54] for the panoramic images of the VIGOR dataset [15]. Our feature extractor for both ground and satellite images is a DINOv2 backbone [49], initialized with FiT weights [48], which is subsequently fine-tuned using an attached DPT module [50]. This extractor yields satellite feature maps with dimensions $(C, H, W) = (32, 128, 128)$. For ground images, initial feature maps of $(32, 64, 256)$ are extracted; these are then projected into BEV using our feature Gaussians projection, resulting in final ground BEV features also of dimensions $(32, 128, 128)$. Our model is trained using the AdamW optimizer [55] (weight decay $10^{-3}$) and a OneCycleLR scheduler [56] with a cosine annealing strategy, where the learning rate peaks at $6.25 \times 10^{-5}$. We use a batch size of 8 on a single 4090 GPU, training for 8 epochs on KITTI and 14cf epochs on VIGOR.

## 4.1 Comparison with State-of-the-Art Methods

We compare our method with the latest state-of-the-art (SOTA) approaches, including supervised methods such as Boosting [4], VFA [20], CCVPE [57], HC-Net [6], and DenseFlow [18], all of which rely on ground-truth camera poses for supervision. We also compare with G2Sweakly [1], which uses only a satellite image and a corresponding ground image as input, similar to our setup.

Table 1: Comparison with the most recent state-of-the-art (SOTA) on KITTI (* denotes supervised learning algorithms). Our weakly supervised approach slightly outperforms SOTA supervised methods in Cross-Area evaluations.

| Algorithms | $\lambda_1$ | Test Area | localization | | Lateral | | Longitudinal | | Azimuth | | | |
|---|---|---|---|---|---|---|---|---|---|---|---|---|
| | | | mean(m)↓ | median(m)↓ | d=1m↑ | d=3m↑ | d=1m↑ | d=3m↑ | $\theta = 1°$↑ | $\theta = 3°$↑ | mean(°)↓ | median(°)↓ |
| Boosting [4]* | - | Same-Area | 12.08 | 11.42 | 76.44 | 96.34 | 23.54 | 50.57 | 99.10 | **100.00** | - | - |
| VFA [20]* | - | | 10.74 | 10.51 | 51.17 | - | 5.19 | - | 49.85 | 96.98 | 1.40 | 1.00 |
| CCVPE [57]* | - | | 1.22 | 0.62 | 97.35 | **98.65** | 77.13 | **96.08** | 77.39 | 99.47 | 0.67 | 0.54 |
| HC-Net [6]* | - | | **0.80** | 0.50 | **99.01** | - | **92.20** | - | 91.35 | 99.84 | 0.45 | 0.33 |
| DenseFlow [18]* | - | | 1.48 | **0.47** | 95.47 | - | 87.89 | - | 89.40 | - | 0.49 | 0.30 |
| G2SWeakly [1] | 0 | | 12.03 | 8.10 | 59.58 | 85.74 | 11.37 | 31.94 | **99.99** | **100.00** | **0.33** | **0.28** |
| Ours | 0 | | 5.82 | 2.85 | 60.04 | 91.54 | 24.06 | 56.82 | **99.99** | **100.00** | **0.33** | **0.28** |
| G2SWeakly [1] | 1 | | 6.81 | 3.39 | 66.07 | 94.22 | 16.51 | 49.96 | **99.99** | **100.00** | **0.33** | **0.28** |
| Ours | 1 | | 2.87 | 2.06 | 52.90 | 94.24 | 35.62 | 76.57 | **99.99** | **100.00** | **0.33** | **0.28** |
| Boosting [4]* | - | Cross-Area | 12.58 | 12.11 | 57.72 | 86.77 | 14.15 | 34.59 | 98.98 | **100.00** | - | - |
| VFA [20]* | - | | 11.12 | 10.95 | 27.82 | - | 5.75 | - | 18.42 | 71.00 | 3.95 | 3.03 |
| CCVPE [57]* | - | | 9.16 | 3.33 | 44.06 | 81.72 | 23.08 | 52.85 | 57.72 | 92.34 | 1.55 | 0.84 |
| HC-Net [6]* | - | | 8.47 | 4.57 | **75.00** | - | **58.93** | - | 33.58 | 83.78 | 3.22 | 1.63 |
| DenseFlow [18]* | - | | 7.97 | 3.52 | 54.19 | - | 23.10 | - | 43.44 | - | 2.17 | 1.21 |
| G2SWeakly [1] | 0 | | 13.87 | 10.24 | 62.73 | 86.53 | 9.98 | 29.67 | **99.99** | **100.00** | **0.33** | **0.28** |
| Ours | 0 | | 7.05 | 3.22 | 58.15 | 92.62 | 23.08 | 51.61 | **99.99** | **100.00** | **0.33** | **0.28** |
| G2SWeakly [1] | 1 | | 12.15 | 7.16 | 64.74 | 86.18 | 11.81 | 34.77 | **99.99** | **100.00** | **0.33** | **0.28** |
| Ours | 1 | | **6.20** | **2.51** | 51.45 | **95.17** | 27.41 | **60.45** | **99.99** | **100.00** | **0.33** | **0.28** |

**KITTI.** The comparison results on the KITTI dataset [51] are summarized in Table 1. Since our rotation estimator is inherited from G2Sweakly [1], the rotation estimation performance is identical between the two methods. However, our method significantly outperforms G2Sweakly [1] in terms of location estimation across almost all evaluation metrics, yielding substantial improvements in both longitudinal pose accuracy and the corresponding mean and median errors. This improvement can be attributed to the limitations of the IPM projection method used in G2Sweakly [1], which suffers from distortions in scenes that are far from the camera and fails to capture the details of objects above the ground plane.

Our feature-based Gaussian splatting for BEV synthesis effectively addresses these issues, leading to a notable enhancement in localization accuracy. Fig. 1 and Fig. 4 visualize the difference between the IPM projection and our proposed BEV synthesis method, clearly demonstrating that our projection technique resolves challenges such as occlusions caused by tall objects (e.g., buildings, trees, vehicles) and geometric distortions from curved roads. Furthermore, in cross-area evaluations, our method even surpasses supervised approaches (Boosting [4], VFA [20], CCVPE [57], HC-Net [6], and DenseFlow [18]) in terms of mean and median errors, showcasing the strong generalization ability of our approach and highlighting the potential of weakly supervised methods.

It is worth noting that the experimental results for VFA [20] are taken from the PIDLoc [22] paper. This is because the authors of VFA have adopted a setting in their prior works [58, 59] that aligns ground-truth poses to the satellite image center, which risks overfitting by biasing predictions towards the center. Since the official code for VFA [20] is not available, we attempted to reproduce their results and found that we could only achieve the reported performance by using this same setting. However, when we use the standard setting adopted by our method and other comparable works [4, 57, 6, 18, 1], our reproduced results are consistent with the VFA results reported in PIDLoc [22]. Therefore, for a fair comparison, we use the VFA [20] results from PIDLoc [22].

**VIGOR.** The comparison results on the VIGOR dataset [15] are presented in Table 2. Our method demonstrates a significant reduction in both mean and median errors compared to the baseline weakly supervised approach, G2SWeakly [1], across all evaluation scenarios. Furthermore, even when benchmarked against state-of-the-art fully supervised methods, our method maintains comparable performance in same-area evaluations while achieving notable improvements across most metrics in cross-area evaluations. This reduces the gap between weakly supervised and fully supervised methods, indicating that our approach generalizes effectively to diverse localization tasks, including both same-area and cross-area scenarios, as well as cases where the query images are either panoramic or captured using pinhole cameras.

Table 2: Comparison with the most recent state-of-the-art (SOTA) on VIGOR (* denotes supervised learning algorithms). Our weakly supervised approach achieves performance comparable to SOTA supervised methods in Same-Area evaluations and comprehensively surpasses them in Cross-Area evaluations.

| Method | $\lambda_1$ | Same-Area | | | | Cross-Area | | | |
| | | Aligned-orientation | | Unknown-orientation | | Aligned-orientation | | Unknown-orientation | |
| | | Mean(m)↓ | Median(m)↓ | Mean(m)↓ | Median(m)↓ | Mean(m)↓ | Median(m)↓ | Mean(m)↓ | median(m)↓ |
|---|---|---|---|---|---|---|---|---|---|
| Boosting [4]* | - | 4.12 | 1.34 | - | - | 5.16 | 1.40 | - | - |
| CCVPE [57]* | - | 3.60 | 1.36 | 3.74 | 1.42 | 4.97 | 1.68 | 5.41 | 1.89 |
| HC-Net [6]* | - | **2.65** | 1.17 | - | - | 3.35 | 1.59 | - | - |
| DenseFlow [18]* | - | 3.03 | **0.97** | 4.97 | 1.90 | 5.01 | 2.42 | 7.67 | 3.67 |
| G2SWeakly [1] | 0 | 5.22 | 1.97 | 5.33 | 2.09 | 5.37 | 1.93 | 5.37 | 1.93 |
| Ours | 0 | 3.15 | 1.45 | 3.18 | 1.49 | 3.03 | 1.41 | 3.05 | 1.43 |
| G2SWeakly [1] | 1 | 4.19 | 1.68 | 4.18 | 1.66 | 4.70 | 1.68 | 4.52 | 1.65 |
| Ours | 1 | 2.87 | 1.58 | **2.91** | **1.60** | **2.84** | **1.36** | **2.89** | **1.38** |

**Computation comparison.** All evaluations of GPU memory usage were performed on an NVIDIA RTX 4090. Our model, which features a DINOv2 [49] backbone fine-tuned with our lightweight DPT network [50] (composed of a few CNN layers), requires considerably less memory during the training phase compared to G2SWeakly [1]. On the KITTI [51] dataset (batch size 8), our training phase uses only 9.2 GB of GPU memory, substantially less than the 22.7 GB required by G2SWeakly [1]. During inference, our model consumes 7.7 GB of GPU memory, while G2SWeakly [1] requires 7.2 GB as shown in Table 5. This increase 0.5 GB for our method is attributable to the larger DINOv2 backbone [49], representing a trade-off for its enhanced feature representation capabilities.

Table 3: BEV synthesis comparison on the KITTI dataset.

| Rendering Method | $\lambda_1$ | Same Area | | Cross Area | |
| | | Mean (m)↓ | Median (m)↓ | Mean (m)↓ | Median (m)↓ |
|---|---|---|---|---|---|
| IPM | 0 | 9.02 | 5.54 | 9.97 | 6.35 |
| Lift-Splat-Shoot [60] | 1 | 16.14 | 13.94 | 17.74 | 14.51 |
| OrienterNet [5] | 0 | 15.59 | 13.68 | 16.15 | 13.80 |
| Direct Projection | 0 | 7.59 | 4.25 | 8.93 | 5.81 |
| BevSplat (w/o OPT) | 0 | 7.42 | 4.16 | 8.81 | 5.74 |
| BevSplat (w/ OPT) | 0 | **5.82** | **2.85** | **7.05** | **3.22** |
| IPM | 1 | 6.68 | 3.71 | 8.60 | 4.84 |
| Lift-Splat-Shoot [60] | 1 | 7.89 | 4.30 | 11.63 | 5.31 |
| OrienterNet [5] | 1 | 5.71 | 3.20 | 10.02 | 5.07 |
| Direct Projection | 1 | 7.59 | 4.25 | 8.93 | 5.81 |
| BevSplat (w/o OPT) | 1 | 4.37 | 3.21 | 7.86 | 4.57 |
| BevSplat (w/ OPT) | 1 | **2.87** | **2.06** | **6.20** | **2.51** |

Table 4: Ablation study on backbone module on the KITTI dataset.

| Methods | $\lambda_1$ | Same Area | | Cross Area | |
| | | Mean(m)↓ | Median(m)↓ | Mean(m)↓ | Median(m)↓ |
|---|---|---|---|---|---|
| Direct Train | 0 | 17.74 | 15.61 | 17.59 | 15.71 |
| LoRA [61] | 0 | 16.29 | 14.48 | 17.05 | 14.7 |
| DPT | 0 | 5.82 | 2.85 | 7.05 | 3.22 |
| Direct Train | 1 | 14.32 | 12.43 | 17.28 | 15.14 |
| LoRA [61] | 1 | 13.58 | 11.79 | 16.81 | 14.63 |
| DPT | 1 | 2.87 | 2.06 | 6.20 | 2.51 |

Table 5: Comparison of resource consumption.

| Method | Training Memory | Inference Memory | Inference Time |
|---|---|---|---|
| OrienterNet [5] | 32.4 | 10.8 | 71 |
| LSS [60] | 26.1 | 8.3 | 85 |
| G2SWeakly [1] | 22.7 | 7.2 | 31 |
| Ours | 9.2 | 7.7 | 44 |

## 4.2 Ablation Study

**Different BEV synthesis approaches.** To validate the effectiveness of our BevSplat method, we compared it against two common BEV generation techniques: the Inverse Perspective Mapping (IPM) approach as utilized in [1], and direct projection of 3D point clouds. For a fair comparison, both baseline methods and our BevSplat employed the same DINOv2 backbone for feature extraction.

*The IPM projection method.* This technique assumes all pixels in the ground-level image

Table 6: Ablation study on backbone module on the KITTI dataset.

| BackBone | $\lambda_1$ | Same Area | | Cross Area | |
| | | Mean(m)↓ | Median(m)↓ | Mean(m)↓ | Median(m)↓ |
|---|---|---|---|---|---|
| VGG | 0 | 8.77 | 4.53 | 9.91 | 5.80 |
| DINOv1 | 0 | 7.68 | 4.01 | 9.16 | 4.67 |
| DINOv2 | 0 | 7.04 | 3.45 | 8.37 | 4.21 |
| DINOv2(FIT) | 0 | **5.82** | **2.85** | **7.05** | **3.22** |
| VGG | 1 | 6.49 | 2.72 | 8.02 | 4.29 |
| DINOv1 | 1 | 4.77 | 2.98 | 7.12 | 3.37 |
| DINOv2 | 1 | 4.21 | 2.73 | 7.01 | 3.18 |
| DINOv2(FIT) | 1 | **2.87** | **2.06** | **6.20** | **2.51** |

correspond to a flat plane at a real-world height of 0 meters. Consequently, while IPM can accurately represent flat road surfaces in BEV, it introduces significant distortions for any objects with non-zero elevations. These objects are typically stretched along the line of sight in the BEV. For instance, in the first row depicted in Fig. 4(b), vehicles' features appear elongated into regions not corresponding to their actual ground footprint. Similarly, in the second row Fig. 4(b), buildings' features are distorted and erroneously projected. A further limitation is that IPM typically projects only the lower portion of the ground image to BEV, discarding valuable information from the upper half. In contrast,

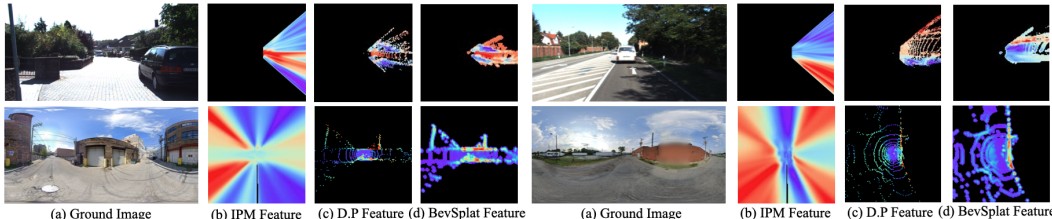

| (a) Ground Image | (b) IPM Feature | (c) D.P Feature | (d) BevSplat Feature | (a) Ground Image | (b) IPM Feature | (c) D.P Feature | (d) BevSplat Feature |

Figure 4: Visualization of query ground images (a), the corresponding BEV feature maps synthesized by IPM (b), by direct projection (c), and by the proposed BevSplat (d). The top two examples are from the KITTI dataset, while the bottom two are from the VIGOR dataset.

our BevSplat method is designed to leverage geometric information from the entire image more effectively.

*Direct point cloud projection to BEV.* While this approach can address some of the aforementioned IPM limitations, it introduces new challenges stemming from point cloud sparsity (visualized in Fig. 4(c)). This sparsity leads to BEV voids and discontinuous features, exacerbated by the lack of control over individual point attributes such as opacity, scale, and shape. Furthermore, unlike 3DGS [7] which utilizes $\alpha$-blending, the simple top-down projection inherent in this method causes severe occlusion, leading to the loss of underlying feature information—an issue also evident in Fig. 4(c). Our ablation study (Table 3) confirms this inherent characteristic: BevSplat(w/o OPT) configured with non-optimizable Gaussian parameters (e.g., fixed opacity=1.0, scale=0.1, offsets=0, and no learned adjustments) performs comparably to direct point cloud projection, underscoring that point clouds can be seen as a degenerate form of 3DGS [7]. However, our full BevSplat(w/ OPT) formulation significantly improves upon this baseline by optimizing Gaussian opacity, scale, shape, and position. Guided by satellite imagery, this optimization process effectively mitigates the issues of point cloud projection, yielding coherent, feature-rich BEV representations and thereby enabling superior localization accuracy. As detailed in Table 3, BevSplat subsequently outperforms both IPM and Direct Projection methods.

*Other depth-based re-sampling methods.* We further compare our method with other BEV projection techniques that are also based on depth prediction, such as those used in Lift-Splat-Shoot [60] and OrienterNet [5] as detailed in Table 3, by adapting their projection modules into our framework. Although these approaches can generate a denser BEV and, like our method, leverage the full vertical information from the ground-view image—allowing them to perform well in same-area settings when guided by GPS labels—they fail to generalize to unseen environments. They tend to make erroneous guesses to fill in occluded regions, which explains their performance degradation in cross-area evaluations. Furthermore, they employ a simple weighted averaging for BEV projection, which is less accurate for handling vertical occlusions compared to BevSplat's principled alpha blending. Finally, both methods utilize a complex $h \times w \times d$ depth representation to perform an attention-based sum over ground features. This implicit, high-dimensional process incurs substantial computational and memory overhead to produce a denser BEV as shown in Table 5.

**Foundation model backbone.** To validate the effectiveness of fine-tuning a foundation model for extracting ground and satellite image features, we conducted ablation studies on the impact of different foundation models with their pre-trained weights, as well as the influence of various fine-tuning methods on the experimental results.

*Impact of Different Foundation Models and Weights.* Prioritizing robust outdoor generalization and effective 3D-relevant feature extraction for our foundation model, we selected DINOv2 [49] fine-tuned with the FiT method [48]. This model, which we term DINOv2(FiT), utilizes its *dinov2_base_fine* pre-trained weights renowned for these capabilities. To validate this choice and compare its efficacy against alternatives, our ablation study also evaluated VGG [62], DINOv1 [63], and the original DINOv2. All DINO-based backbones in this study (DINOv1, DINOv2, and DINOv2(FiT)) were subsequently further fine-tuned by us using a DPT-like module [50]. The ablation results (Table 6) validated our selection, as DINOv2(FiT), after our DPT-like fine-tuning, demonstrated superior performance among the evaluated backbones.

*Impact of Different Fine-tuning Methods.* Using DPT-like models is a common practice for 3D vision tasks; for example, VGGT [64] utilizes DPT [50] for point cloud reconstruction, depth estimation, and

feature matching. Although we are the first to apply DPT-DINO for feature extraction in the specific sub-field of cross-view localization, our motivation is to similarly obtain features that are rich in 3D information. This is analogous to human navigation, where in addition to semantic information, an understanding of the real 3D scene is also crucial for localization. However, obtaining such 3D-aware features to bridge the significant ground-satellite domain gap is non-trivial. Simpler fine-tuning methods like direct end-to-end training or LoRA [61] fail, as they either lose crucial texture details or are not powerful enough to adapt the foundation model. We use a DPT-like module because its multi-scale feature fusion architecture is uniquely suited for this challenge. It successfully adapts the backbone by preserving both the low-level texture and high-level semantic information required for matching across these different domains as shown in Table 4.

**Number of Gaussian primitives per pixel** ($N_p$). The number of Gaussian primitives per pixel, $N_p$, also affects the resulting BEV feature quality. While an excessive $N_p$ can complicate training and cause inter-primitive occlusions, an insufficient count leads to sparse and inadequate BEV representations. Our ablation study (Fig. 5) determined $N_p = 3$ to be optimal, offering the best balance between feature richness and model tractability.

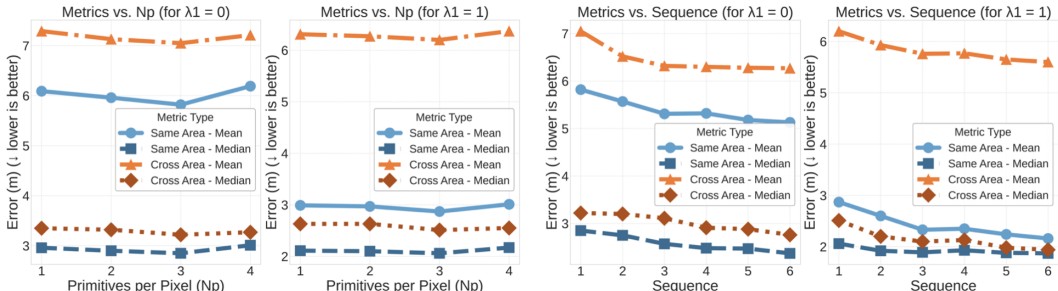

Figure 5: Ablation on primitives per pixel ($N_p$) The error is minimized when $N_p = 3$ on KITTI dataset.

Figure 6: Location error with increasing sequence length on KITTI dataset.

### 4.3 Multi-Frame Localization

Beyond processing single ground-level images, our method extends to leveraging multiple frames from video sequences to enhance localization robustness, particularly in dynamic environments. Similar to CVLNet [65], given known inter-frame relative poses, our BevSplat technique projects features from these frames into a unified BEV. These multi-frame BEV features are then fused by a Transformer employing self-attention. We investigated this multi-frame capability using query video sequences comprising 1 to 6 frames. The results, presented in Fig. 6, demonstrate that localization performance consistently improves with an increasing number of frames in the sequence. This underscores the efficacy of our approach in leveraging temporal information from video data for enhanced localization robustness.

## 5 Conclusion

This paper has introduced a new approach for weakly supervised cross-view localization by leveraging feature-based 3D Gaussian primitives to address the challenge of height ambiguity. Unlike traditional methods that assume a flat ground plane or rely on computationally expensive models such as cross-view transformers, our method synthesizes a Bird's-Eye View (BEV) feature map using feature-based Gaussian splatting, enabling more accurate alignment between ground-level and satellite images. We have validated our approach on the KITTI and VIGOR datasets, demonstrating that our model achieves superior localization accuracy.

However, the inference speed of our method is currently constrained by the reconstruction and rendering overhead inherent to existing 3D Gaussian Splatting (3DGS) techniques. Future work will focus on developing faster reconstruction algorithms and more compact 3D Gaussian representations to enhance computational efficiency. Despite this current limitation, we believe that our approach provides a promising direction for scalable and accurate cross-view localization, paving the way for real-world applications in autonomous navigation, geospatial analysis, and beyond.

## Acknowledge

The authors are grateful for the valuable comments and suggestions by the reviewers and ACs. This work was supported by NSFC (62406194), Shanghai Frontiers Science Center of Human-centered Artificial Intelligence (ShangHAI), MoE Key Laboratory of Intelligent Perception, HPC Platform of ShanghaiTech University and Human-Machine Collaboration (KLIP-HuMaCo). A part of the experiments of this work were supported by the core facility Platform of Computer Science and Communication, SIST, ShanghaiTech University.

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

## A    Carefule Explanation of the Weakly Supervised Setup

We focus on a weakly-supervised setting because precise GPS data is often unavailable in the real world. To do this, we adopt the weakly-supervised setup from G2SWeakly [1], which defines two scenarios:

$\lambda=0$: the error of the location labels for ground images in the training dataset is the same as the error that the model aims to refine during deployment. For example, the error of location labels for ground images in the training data set is +/- 20m. During testing, the model is also given a location of query images with error up to 20m and aim to reduce this error.

$\lambda=1$: relatively more accurate location labels for ground images in the training data are available than the poses we aim to refine during employment. For example, the model was trained with images whose location labels have an error of +/- 5m. During testing, the query images have an initial location estimate with errors up to 20m, and the model aim to reduce this error.

## B    Concepts of "Height Ambiguity".

We use the term "height ambiguity" to describe the challenge of projecting a 2D ground-level image to a Bird's-Eye View (BEV). Because a single 2D pixel could represent points at various depths and heights, its true 3D position is ambiguous.

Methods like IPM resolve this ambiguity by assuming a flat ground plane. This introduces significant errors for any object with height, causing the characteristic distortions and smearing we aim to solve. We will clarify this definition in our revision

## C    Clarification on Occlusion Handling in the Differentiable Blending Process

**Forward Pass:** Following Equation 3, we sort all contributing Gaussians by depth and render them from front to back.

**Backward Pass:** The process is fully differentiable, allowing the loss to guide how the scene should be structured. For the feature vector ($f_b$): The gradient for the $b$-th Gaussian is computed as:

$$\frac{\partial L_{\text{all}}}{\partial f_b} = \frac{\partial L_{\text{all}}}{\partial F_{\text{BEV}}} \cdot T_b \cdot \alpha_b \tag{8}$$

The learning signal is scaled by transmittance ($T_b$) and opacity ($\alpha_b$). This means the features of the most visible (least occluded) and most solid Gaussians are prioritized for updates.

For the opacity ($\alpha_b$): The gradient of the $b$-th Gaussian is computed as:

$$\frac{\partial L_{\text{all}}}{\partial \alpha_b} = \frac{\partial L_{\text{all}}}{\partial F_{\text{BEV}}} \cdot T_b \cdot (f_b - f_b^{\text{accum}}) \tag{9}$$

The gradient depends on the difference between the current Gaussian's feature ($f_b$) and the accumulated features behind it ($f_b^{\text{accum}}$). This trains the model to make a Gaussian opaque if it is needed to hide a conflicting background, effectively learning to form solid, occluding surfaces.

In short, this mechanism is directly analogous to how the original 3DGS handles RGB colors, and we have repurposed it to optimize feature representations for localization.

## D    Robustness to Localization Errors

We evaluate the robustness of our method to varying levels of initial localization error. As shown in Table 7, localization performance improves significantly as the initialization error decreases.

## E    Ablation Study on Gaussian Primitive Offset and Scale

This ablation study investigates our method's sensitivity to the maximum offset and maximum scale of Gaussian Primitives. For each parameter, we evaluate values from the set {0.3, 0.5, 1.0}. The results,

Table 7: Performance comparison under different location error settings on KITTI dataset.

| Location Error (m²) | $\lambda_1$ | Same Area | | Cross Area | |
|---|---|---|---|---|---|
| | | Mean(m) ↓ | Median(m) ↓ | Mean(m) ↓ | Median(m) ↓ |
| $56 \times 56$ | 0 | 5.82 | 2.85 | 7.05 | 3.22 |
| | 1 | 2.87 | 2.06 | 6.20 | 2.51 |
| $28 \times 28$ | 0 | 3.27 | 2.28 | 3.60 | 2.47 |
| | 1 | 2.43 | 1.94 | 3.31 | 2.21 |

presented in Table 8, demonstrate relatively stable performance across these configurations. Optimal performance is observed when both the maximum offset and scale are set to 0.5; consequently, these are adopted as their default values.

Table 8: Ablation study on max_offset and max_scaleon KITTI dataset.

| Max_Offset(m) | Max_Scale(m) | $\lambda_1$ | Same Area | | Cross Area | |
|---|---|---|---|---|---|---|
| | | | Mean(m) ↓ | Median(m) ↓ | Mean(m) ↓ | Median(m) ↓ |
| 0.3 | 0.3 | 0 | 6.16 | 2.89 | 7.36 | 3.20 |
| 0.5 | 0.5 | 0 | **5.82** | **2.85** | **7.05** | 3.22 |
| 1.0 | 1 | 0 | 6.00 | 2.95 | 7.06 | **3.16** |
| 0.3 | 0.3 | 1 | 3.42 | 2.28 | 6.83 | 2.53 |
| 0.5 | 0.5 | 1 | **2.87** | **2.06** | **6.20** | **2.51** |
| 1.0 | 1 | 1 | 3.28 | 2.30 | 6.52 | 2.57 |

# F  Ablation Study on the Number of Gaussian Primitives Per Pixel($N_p$)

We conducted an ablation study on the number of Gaussian primitives per pixel ($N_p$) across both the KITTI and VIGOR datasets to validate our design choice. The results for KITTI are presented in Figure 5 of our main paper, and the new results for the VIGOR dataset are provided below Table 9.

Table 9: Ablation study on the number of sampled points, $N_p$.

| $N_p$ | $\lambda_1$ | Same Area | | Cross Area | |
|---|---|---|---|---|---|
| | | Mean(m) ↓ | Median(m) ↓ | Mean(m) ↓ | Median(m) ↓ |
| 1 | 0 | 3.05 | 1.71 | 2.97 | 1.71 |
| 2 | 0 | 2.98 | 1.67 | 2.94 | 1.65 |
| 3 | 0 | **2.96** | **1.62** | **2.90** | **1.65** |
| 4 | 0 | 3.03 | 1.67 | 2.91 | 1.68 |
| 1 | 1 | 2.62 | 1.47 | 2.71 | 1.42 |
| 2 | 1 | 2.59 | 1.41 | 2.67 | 1.40 |
| 3 | 1 | **2.57** | **1.40** | **2.63** | **2.54** |
| 4 | 1 | 2.59 | 1.42 | 2.65 | 1.38 |

The results on VIGOR are consistent with our findings on KITTI: performance is optimal when =3. As we discuss in our paper:

- Using too few primitives can limit the model's ability to fill gaps in sparse regions.
- Using too many primitives can make training more difficult.

This finding is also consistent with prior work. Our design was inspired by PixelSplat [66], which similarly found Np=3 to be a robust and effective setting across multiple datasets. Therefore, we conclude that Np=3 is a well-justified hyperparameter that should be generally applicable.

## G   Applicability to Multi-Frame Localization Tasks

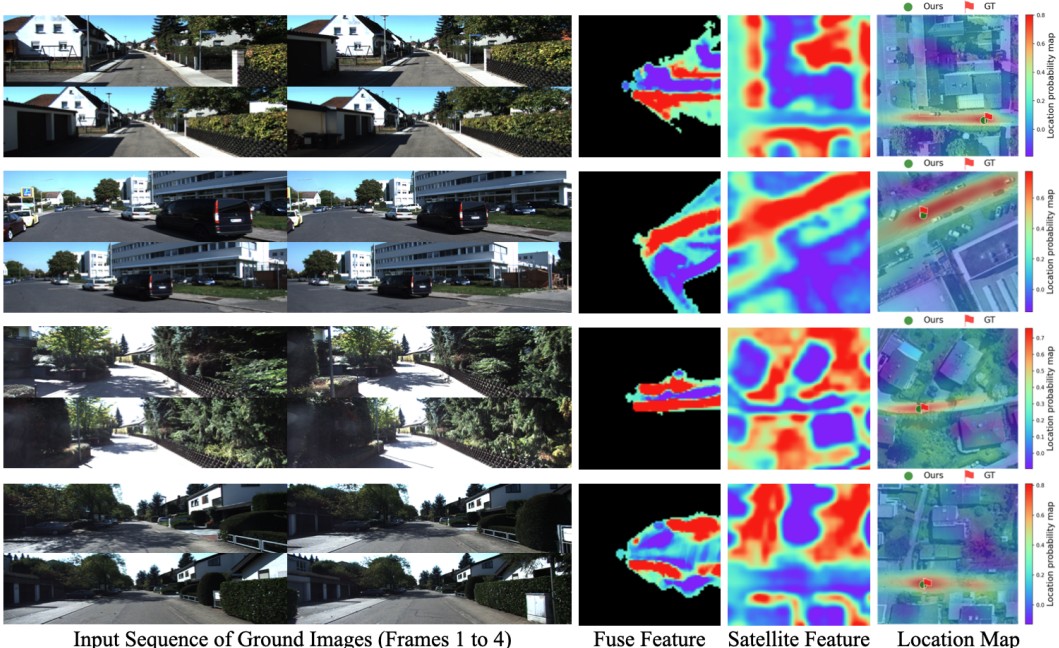

Input Sequence of Ground Images (Frames 1 to 4)  Fuse Feature  Satellite Feature  Location Map

Figure 7: Visualization of multi-frame localization results on the KITTI dataset, achieved using our BevSplat-based approach. This demonstrates the aggregation of information over time and the filtering of dynamic elements.

As detailed in the main paper, our quantitative analysis of a CVLNet-based multi-frame fusion method [65] demonstrated progressively improved performance with an increasing number of frames, confirming its efficacy for temporal sequence tasks. To complement these findings, Figure 7 provides a qualitative illustration.

This visualization highlights how our fusion strategy effectively aggregates richer contextual information from multiple video frames. Notably, the approach adeptly filters dynamic objects while prioritizing the preservation of static scene elements, which are crucial for robust cross-view localization. These qualitative insights further substantiate the effectiveness and generalizability of our proposed method in handling dynamic environments and leveraging temporal information for improved localization.

## H   Multi-frame Fusion Comparison: BevSplat vs. IPM and Direct Point Cloud Projection

To demonstrate our method's consistency and fusion capabilities, we have conducted a new multi-frame comparison against both IPM and direct point cloud projection baselines. The Table 10 below present the performance on the KITTI dataset using a sequence of six frames. The values in parentheses show the percentage improvement from fusing six frames over the single-frame results:

Our BevSplat framework demonstrates superior multi-frame fusion capabilities for the following reasons:

- Inverse Perspective Mapping (IPM): BEV representations generated via IPM are prone to significant artifacts and distortions, which fundamentally hinder effective temporal fusion.

- Direct Point Cloud: Projections of raw point clouds result in sparse representations that handle occlusions poorly, often causing background features to bleed through foreground objects. This issue is not resolved well by aggregating multiple frames.

Table 10: Mutil-frame fusion comparison.

| Methods | Seq | $\lambda_1$ | Same Area | | Cross Area | |
|---|---|---|---|---|---|---|
| | | | Mean(m) | Median(m) | Mean(m) | Median(m) |
| G2SWeakly | 6 | 0 | 8.65($\downarrow$4.1%) | 5.22($\downarrow$5.7%) | 9.41($\downarrow$4.6%) | 6.01($\downarrow$5.3%) |
| Direct Projection | 6 | 0 | 7.05($\downarrow$7.1%) | 3.86($\downarrow$9.2%) | 8.15($\downarrow$8.7%) | 5.31($\downarrow$8.6%) |
| Ours | 6 | 0 | 5.01($\downarrow$13.9%) | 2.27($\downarrow$20.4%) | 6.09($\downarrow$13.6%) | 2.71($\downarrow$15.8%) |
| G2SWeakly | 6 | 1 | 6.25($\downarrow$6.4%) | 3.38($\downarrow$8.9%) | 8.18($\downarrow$4.9%) | 4.32($\downarrow$10.7%) |
| Direct Projection | 6 | 1 | 3.85($\downarrow$13.1%) | 2.91($\downarrow$11.6%) | 7.18($\downarrow$9.6%) | 3.96($\downarrow$14.7%) |
| Ours | 6 | 1 | 2.01($\downarrow$30.0%) | 1.77($\downarrow$14.1%) | 5.23($\downarrow$15.6%) | 1.94($\downarrow$22.7%) |

- **Our Method:** In contrast, BevSplat utilizes adaptive Gaussian primitives to create a dense, coherent BEV that faithfully represents the road topology. This provides a robust foundation for multi-frame fusion, as shown in Fig. 9 of our supplement.

## I  Sensitivity to Depth Prediction Quality and Failure Cases

We evaluate our framework's performance with three different depth foundation models: DepthAnythingv1 [67], ZoeDepth [68], and DepthAnythingv2 [69] in Table 11:

Table 11: Ablation study on different depth estimation methods.

| Method | $\lambda_1$ | Same Area | | Cross Area | |
|---|---|---|---|---|---|
| | | Mean(m)$\downarrow$ | Median(m)$\downarrow$ | Mean(m)$\downarrow$ | Median(m)$\downarrow$ |
| DepthAnythingV1 [67] | 0 | 5.91 | 2.84 | 7.21 | 3.25 |
| ZoeDepth [68] | 0 | 5.84 | 2.86 | 7.14 | 3.22 |
| DepthAnythingV2 [69] | 0 | 5.82 | 2.85 | 7.05 | 3.22 |
| DepthAnythingV1 [67] | 1 | 2.97 | 2.11 | 6.28 | 2.52 |
| ZoeDepth [68] | 1 | 2.91 | 2.03 | 6.21 | 2.54 |
| DepthAnythingV2 [69] | 1 | 2.87 | 2.06 | 6.20 | 2.51 |

The results demonstrate that while a more accurate depth model improves performance, the overall system is not highly sensitive to the choice of different depth estimators. This robustness stems from our end-to-end differentiable design, which optimizes the initial 3D Gaussian positions during training, compensating for minor discrepancies between different depth priors. Our framework can seamlessly leverage future advancements in monocular depth estimation. We will include this analysis in our paper.

## J  Robustness in Adverse Environmental Conditions

To test our method's robustness, we generated synthetic Rain, Fog, and Night data for KITTI (following the methodology of Robust-Depth [70]). This allows for a controlled comparison against the G2SWeakly baseline under challenging conditions.

The results in Table 12 show that while both methods are affected by adverse conditions, our approach demonstrates greater relative robustness. The reason lies in how each method handles corrupted input:

- IPM-based methods like G2SWeakly project visual artifacts (e.g., rains, fog) directly onto the BEV, creating severe geometric distortions that corrupt the final representation.
- In contrast, our method, while starting with a less accurate depth map in these conditions, still preserves a stable underlying 3D structure. It avoids the stretching errors of IPM and is better able to ignore atmospheric noise, leading to a more graceful degradation in performance.

Table 12: Performance comparison under different weather conditions.

| Method | Weather | $\lambda_1$ | Same Area | | Cross Area | |
|---|---|---|---|---|---|---|
| | | | Mean(m)↓ | Median(m)↓ | Mean(m)↓ | Median(m)↓ |
| G2SWeakly | Origin | 0 | 9.02 | 5.54 | 13.97 | 10.24 |
| | Rain | 0 | 16.45 | 13.29 | 18.44 | 16.52 |
| | Fog | 0 | 12.82 | 9.8 | 15.47 | 12.03 |
| | Night | 0 | 14.42 | 11.31 | 17.25 | 14.66 |
| Ours | Origin | 0 | 5.82 | 2.85 | 7.05 | 3.22 |
| | Rain | 0 | 8.61 | 4.78 | 10.03 | 5.69 |
| | Fog | 0 | 6.60 | 3.23 | 8.27 | 4.29 |
| | Night | 0 | 7.59 | 4.11 | 10.77 | 6.16 |
| G2SWeakly | Origin | 1 | 6.68 | 3.71 | 12.15 | 7.16 |
| | Rain | 1 | 16.82 | 13.48 | 19.45 | 17.42 |
| | Fog | 1 | 10.19 | 5.48 | 15.67 | 12.53 |
| | Night | 1 | 11.56 | 7.43 | 17.72 | 16.53 |
| Ours | Origin | 1 | 2.87 | 2.06 | 6.20 | 2.51 |
| | Rain | 1 | 8.64 | 3.94 | 11.12 | 6.34 |
| | Fog | 1 | 4.21 | 2.50 | 8.21 | 3.43 |
| | Night | 1 | 7.95 | 3.32 | 10.39 | 7.09 |

## K   Coordinate System

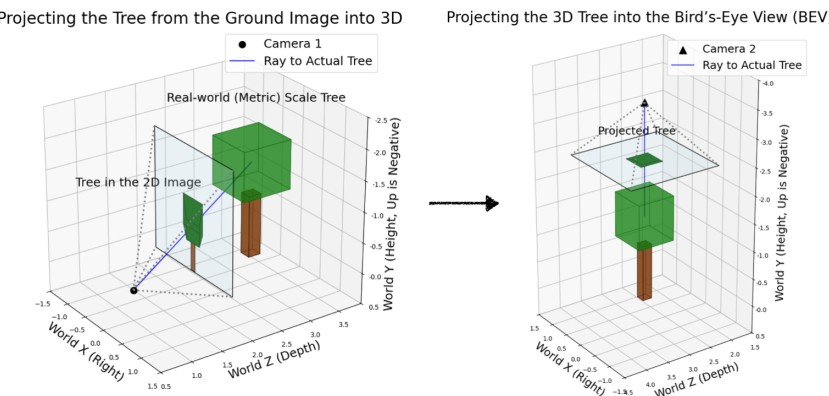

Figure 8: BevSplat Geometry Projection Overview. Our method is a two-stage geometry projection. *Left panel (Stage 1):* We reconstruct the 3D scene from ground-level images using their associated depth information, illustrated by converting a tree from a ground-level image to its 3D representation. *Right panel (Stage 2):* The reconstructed 3D scene is then projected into the Bird's-Eye View (BEV).

Our methodology employs a world coordinate system consistent with the OpenCV convention [71], as depicted in Figure 8. This is a right-handed system where, from the camera's viewpoint, the +X axis extends to its right, the +Y axis points downwards, and the +Z axis aligns with its forward viewing direction. Consequently, the upward direction corresponds to the -Y axis.

In the 3D reconstruction stage, a point cloud is generated from the input images. This is achieved by back-projecting pixels, using their depth information, along the initial camera's viewing direction (defined as the +Z axis of this coordinate system).

Subsequently, for Bird's-Eye View (BEV) projection, an aerial perspective is simulated. A virtual camera is conceptually positioned at a nadir viewpoint—looking directly downwards—above the reconstructed 3D scene. Given our coordinate system where the +Y axis points downwards, this BEV camera is located at a Y-coordinate that is numerically smaller than those of the scene's primary content (thus representing a higher altitude). It views along the +Y direction (downwards). The BEV

is then formed by orthographically projecting the 3D point cloud onto the world's XZ-plane (which effectively serves as the ground plane) along this +Y viewing axis.

## L    Qualitative Results

**Robust Performance in Complex Scenarios:** As illustrated in the first two rows of images in Figure 9 and Figure 10, our method demonstrates robust localization performance across a variety of challenging scenarios, such as road intersections, curved road sections, and areas with significant occlusions from roadside trees, as validated on the KITTI and VIGOR datasets. This proficiency is primarily attributed to our approach's enhanced capabilities in: (1) effectively handling visual occlusions caused by buildings; (2) establishing and leveraging more accurate geometric relationships within the scene; and (3) optimally fusing features pertinent to the vertical spatial arrangement of elements, such as trees and road surfaces, between ground-level and aerial (*e.g.*, satellite) views. Consequently, our method achieves promising localization results in these complex environments, underscoring its effectiveness in tackling real-world complexities.

**Limitations in Feature-Scarce Environments:** Conversely, as illustrated in the last two rows of images in Figure 9 and Figure 10, in specific scenarios such as long, straight road segments that lack distinctive visual features, our method exhibits a comparative reduction in localization accuracy. The primary reason for this limitation is that in the absence of salient visual landmarks, the deep learning network, when attempting to match the ground-level view to the satellite imagery, may assign similar matching probabilities or confidence scores to multiple plausible locations within the satellite map. This multi-modal matching outcome leads to localization ambiguity, making it difficult for the network to make a unique, high-precision positioning decision.

## M    On the Benefits in Supervised vs. Weakly-Supervised Settings

Although our paper focused on the weakly-supervised setting, our framework also demonstrates strong performance with full supervision. To illustrate this, we trained our model and the G2SWeakly baseline in a fully supervised setting on the KITTI dataset. The results are in Table 13:

Table 13: Supervised vs. weakly-supervised settings.

| Method | $\lambda1$ | Same Area | | Cross Area | |
|---|---|---|---|---|---|
| | | Mean(m) $\downarrow$ | Median(m) $\downarrow$ | Mean(m) $\downarrow$ | Median(m) $\downarrow$ |
| G2SWeakly(Supervised) | - | 6.32 | 3.15 | 12.2 | 8.33 |
| Ours(Supervised) | - | 2.07 | 1.12 | 6.75 | 3.03 |
| Ours(Weakly Supervised) | 0 | 5.82 | 2.85 | 7.05 | 3.22 |
| Ours(Weakly Supervised) | 1 | 2.61 | 2.06 | 6.20 | 2.51 |

As the results show, while switching to a fully supervised setting, our model is highly competitive, in the challenging cross-area task. However, we found that our weakly-supervised model ($\lambda_1$=1) achieves even better cross-area performance than our own fully-supervised version. This suggests that precise supervision may cause overfitting to the training domain's biases, which is contrary to our goal of building a more generalizable system.

Therefore, our paper's focus on the weakly-supervised setting is twofold. First, it addresses the practical challenge that high-quality GPS data is often unavailable in the real world. Second, it is the setting where our method paradoxically achieves its best and most robust generalization performance.

## N    Limitations and Future Works

As discussed in the main paper's conclusion, a current limitation of our BevSplat method, which renders Bird's-Eye View (BEV) perspectives based on 3D Gaussian Splatting [7], is its computational speed compared to Inverse Perspective Mapping (IPM). For instance, on an NVIDIA RTX 4090 GPU, BevSplat requires 14 ms to generate a single BEV image. In contrast, IPM, which utilizes

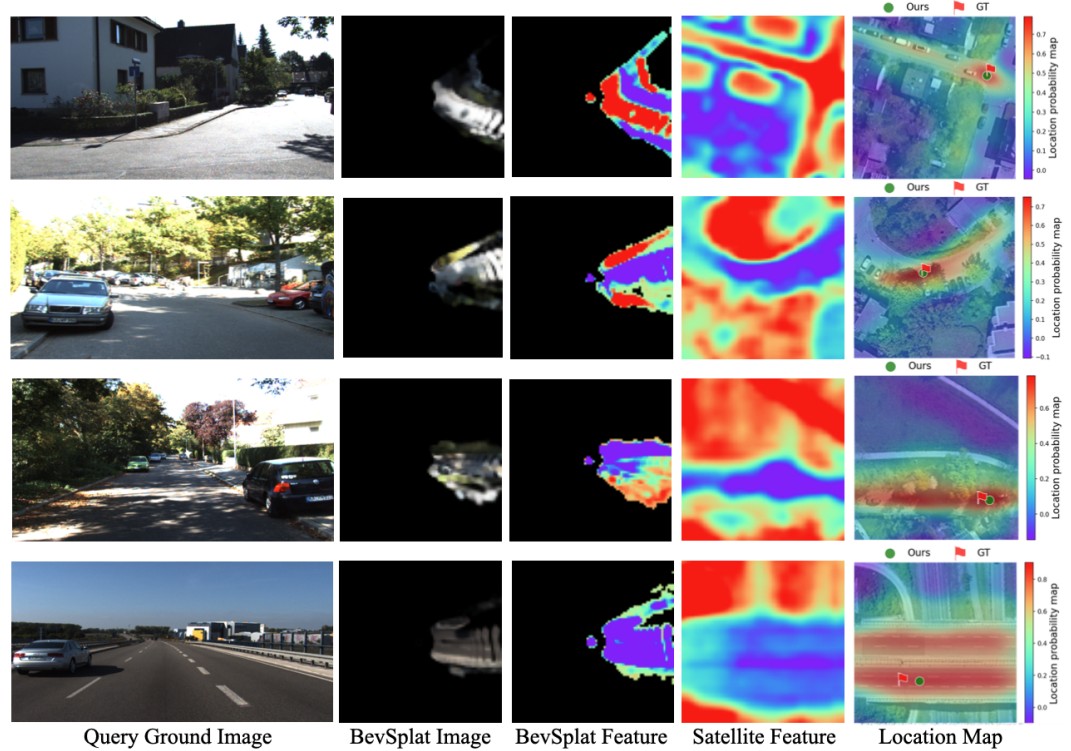

| Query Ground Image | BevSplat Image | BevSplat Feature | Satellite Feature | Location Map |

Figure 9: Qualitative results for BevSplat-based single-image localization on KITTI. Top two rows: successful examples; bottom two rows: failure examples.

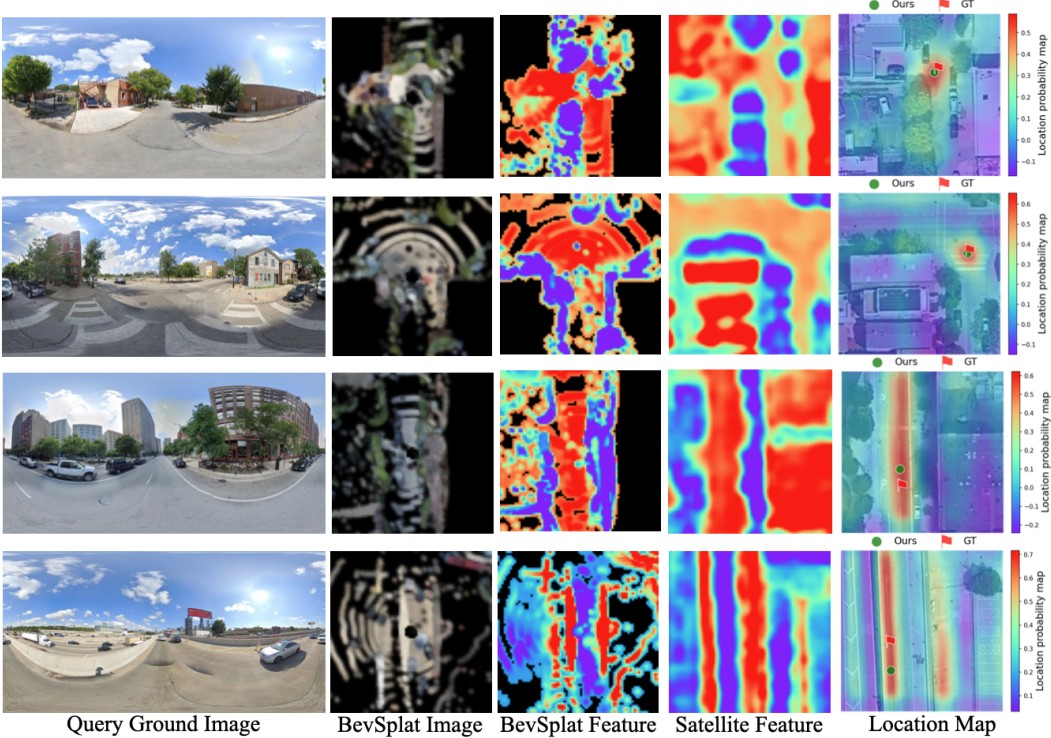

| Query Ground Image | BevSplat Image | BevSplat Feature | Satellite Feature | Location Map |

Figure 10: Qualitative results for BevSplat-based single-image localization on VIGOR. Top two rows: successful examples; bottom two rows: failure examples.

direct linear interpolation, can achieve this in 4ms. This performance disparity affects the overall inference speed of our model. Therefore, a significant direction for our future work is the exploration of faster and more compact Gaussian representations to address this bottleneck and enhance real-time applicability.

## O   Broader Impacts

Our work, BevSplat, addresses the critical demand for robust and accessible localization systems for mobile robots, such as drones and autonomous vehicles, particularly in scenarios where high-precision GPS is either unavailable or impractical due to cost or signal dependency. By leveraging computer vision, BevSplat delivers real-time, high-precision localization using only a monocular camera, or a camera augmented with low-cost, low-precision GPS. This significantly extends localization capabilities to GPS-denied or unreliable environments, a crucial step for the widespread adoption of autonomous systems.

To foster further research and collaboration within the community, we are committed to open-sourcing our complete codebase, training datasets, and pre-trained model weights on GitHub. This efficient implementation, which operates on a single NVIDIA RTX 4090 GPU, is provided as a resource for the research community. We encourage researchers to explore, build upon, and collaborate with us to advance this promising research direction, ultimately contributing to safer and more versatile autonomous navigation.

