# OpenReview forum: "BevSplat: Resolving Height Ambiguity via Feature-Based Gaussian Primitives for Weakly-Supervised Cross-View Localization"
_NeurIPS.cc/2025/Conference — NeurIPS 2025 spotlight_

### Official Review · Reviewer_WBEg · 2025-06-26

**Clarity:** 2
**Significance:** 3
**Originality:** 3
**Rating:** 5
**Confidence:** 3

**Summary:**

The draft describes a system to localize a ground image relative to a satellite image. Building on previous work, G2SWeakly [1], the method can similarly be trained in a weakly supervised fashion, only requiring ground-satellite pairs with coarse GPS labels for the ground images. The main contribution on top of G2SWeakly is to substitute Inverse Perspective Mapping (IPM) with feed-forward prediction and birds-eye-view (BEV) re-rendering of Gaussian Splats. The splats encode high dimensional neural features, such that the BEV re-render is a neural representation of the ground image, which can be matched against a neural map extracted from the satellite image. Experiments are conducted on two datasets, Kitti and VIGOR, and show that the method a) consistently outperforms the baseline G2SWeakly, b) performs on par or even better than supervised competitors.

**Questions:**

- Please discuss the results of [20]. Could these numbers be included in Table 1, or is the comparison unfair for some reason? If these numbers were included, which claims would still hold? If sufficient reason were provided that [20] would not work in a weakly supervised fashion (e.g. by coupling [1] with [20]) I would consider accepting it's superior performance as being entirely limited to the strongly supervised domain, and raise my score.

- Within your framework, would it be possible to adapt the BEV projection approaches of [5] or [20]? If those two baselines would be included in the draft, and the proposed BEV splats projection still shows benefits, I am willing to increase my rating.

**Ethical Concerns:**

["NO or VERY MINOR ethics concerns only"]

**Final Justification:**

The authors have addressed my main concerns in the rebuttal. They provide the missing comparison to the BEV projection of [5] which performs worse than the proposed approach. They also provided a possible explanation of why [5] performs worse.

Regarding the missing discussion of the results of [20], the authors explain that those results might be an artifact of a particular way to evaluate, that seems to re-appear throughout works of the authors of [20]. They substantiate this claim with experiments using a custom implementation, and with pointers to recent related work that similarly call the results of [20] into question.

I encourage the authors to add this discussion and the various experiments to the submission / the supplement. With those changes, the submission represents a solid advance in weakly supervised cross-view relocalization, carried by strong results. The other reviews echo this assessment. Therefore, I recommend to accept the draft.

**Limitations:**

yes (As a minor comment I would suggest to actually provide the "inference speed" of the method rather than only saying it is constrained.)

**Paper Formatting Concerns:**

-

**Quality:**

2

**Strengths And Weaknesses:**

**Neutral: The writing is of mixed quality**. Initially, I had trouble understanding the setup and motivation of the paper, since the writing assumes that the reader is familiar with previous works on this problem.
* Concepts like "height ambiguity" are repeatedly mentioned without being explained.
* The limitations of Inverse Perspective Mapping (IPM) are assumed to be obvious throughout the text, and IPM is only explained on page 8.
* The related work section mentions "one-to-many cross-view localization" or "pixel-level localization" without explaining what it is.
* The technical contributions are not listed clearly, e.g. using a list of bullet points after the introduction. Only after reading the entire draft it becomes apparent that the main contribution is improving on the IPM step in G2SWeakly.
* Parts of the pipeline, stemming from G2SWeakly, are omitted. The experimental section mentions that the proposed method re-uses the rotation estimation step of G2SWeakly. I do not think the method description ever touches on this aspect, I was not even aware that the rotation needs to be estimated in a designated processing step. I imagined rotation is just solved when matching the BEV features to the satellite features. Instead, there seem to be extra parts of the pipeline that are inherited from G2SWeakly but never discussed.
* On the other hand, I found the technical description - of how the Gaussian Splats are predicted and used - well written and easy to follow.
* The figures, e.g. the system overview in Fig. 2., are well done and informative.
* Overall, despite the issues above, after reading the entire text, I had the impression that I understood the method and which problems it solves.

**Strengths:**
* The method advances capabilities of weakly-supervised cross-view localization which is a significant family of methods due to their ability to scale (precise ground-to-satellite pose labels are difficult to get).
* The experimental sections shows consistent and notable improvements over the baseline, G2SWeakly. The method also compares well with fully supervised approaches.
* The draft includes a multi-frame variation of the method which shows how accuracy can be improved if the user provides a short image sequence rather than a single image. Such a setup is of practical value.

**Weaknesses:**
* The draft cites [20] ("View from above", CVPR24) but does not discuss it. That method, while being fully supervised, seems to achieve superior results on the Kitti dataset, affecting some claims of the submission. I.e. claims like "in cross-area evaluations, our method even surpasses supervised approaches" (line 224) do not hold if the results of [20] were included in Table 1. [20] would lead by a large margin. This is made worse because [20] technically seems to address the same problems as the draft: namely incorporating features above the ground plane in ground-to-satellite matching. This needs to be discussed in detail.
* The baselines for BEV projection are weak, and alternative approaches from the literature to do the BEV projection are ignored. The only alternative to naive IPM considered, is a re-rendering of the ground depth map from BEV. This clearly leads to a very sparse BEV representation that is difficult to match with the neural map, as discussed in the draft. But better strategies exist, e.g. the depth-based re-sampling of OrienterNet [5] which leads to a dense BEV or the aforementioned approach of [20].

**Conclusion:**
The main contribution of the draft is to present an alternative to Inverse Perspective Mapping (IPM), but previous strategies, like [5, 20] are not discussed nor compared to. Therefore, important baselines are missing.

---

> ### Author Rebuttal · Authors · 2025-07-31
>
> We thank Reviewer WBEg for the detailed review and constructive suggestions. We address the major concerns below.
> ## 1. Writing
> We sincerely thank you for your careful review and valuable feedback. We would like to provide explanations for the points you raised.
>
>
> ### More Explanation of IPM.
> IPM was defined on page 1, line 32 and visualized in Figure 1 to motivate our work.
> It assumes that all points on the ground plane, and exploit the ground plane Homography for the ground-to-satellite projection.
> We will highlight this more clearly in the revision.
>
> ### Concepts of "Height Ambiguity".
>
> Thank you for the question. We use the term "height ambiguity" to describe the challenge of projecting a 2D ground-level image to a Bird's-Eye View (BEV). Because a single 2D pixel could represent points at various depths and heights, its true 3D position is ambiguous.
>
> Methods like IPM resolve this ambiguity by assuming a flat ground plane. This introduces significant errors for any object with height, causing the characteristic distortions and smearing we aim to solve. We will clarify this definition in our revision
>
>
> ### Different Localization Task Definitions in Related Works
> We will clarify these terms in the paper.
> * One-to-many localization: this task aims to determine the coarse location of a ground camera by retriving its similar satellite image counterparts from the database. The similar satellite image counterparts can be several, and thus is named as one-to-many localization.
> * Pixel-level localization: once the coarse location of a ground camera is given, the pixel-level localization aims to determine which pixel of the satellite image corresponds to the precise ground camera pose. This paper solves for the pixel-level localization.
>
>
> ### Rework the presentation of the paper's contributions.
> Thank you for the suggestion. Our contribution is re-worked below:
>
> * We introduce BevSplat, a new framework that synthesizes a Bird's-Eye View (BEV) representation by modeling the ground scene as a collection of feature-based 3D Gaussian primitives.
> * Our approach explicitly models 3D geometry to resolve the critical issue of height ambiguity, a limitation of traditional projection methods like IPM that leads to severe BEV distortions.
> * We demonstrate that BeVSplat achieves new state-of-the-art performance on the challenging KITTI and VIGOR datasets, significantly outperforming prior methods within the practical weakly-supervised localization paradigm.
>
>
> ### Rotation Estimation
> Yes, our rotation estimation module is adopted directly from G2SWeakly. We will make this explicit in our revised pipeline description.
>
> ## 2. Why not compare with "View from Above", will claims like "in cross-area evaluations, our method even surpasses supervised approaches" (line 224) still hold?
>
> The primary reason we did not include a direct comparison with "View from Above (VFA)" [20] was the unavailability of its official source code and pre-trained models, which prevents a direct and fair reproducible comparison. We note that even the most recent state-of-the-art supervised method, FG<sup>2</sup> [21] (CVPR'25), also omits a comparison to VFA for the same reason.
>
> To address your crucial question about whether our claim on line 224 still holds, we have benchmarked our weakly-supervised method against this new fully-supervised SOTA method, FG²[21]. The results are presented below:
>
>
> | Methods      | λ1 | Same Area mean(m) ↓ | Same Area median(m) ↓ | Cross Area mean(m) ↓ | Cross Area median(m) ↓ |
> |:-------------|:--:|:-------------------:|:---------------------:|:--------------------:|:----------------------:|
> | FG^2[21] | -  |         **0.75**        |          **0.52**         |         7.45         |          4.03          |
> | Ours         | 0  |         5.82        |          2.85         |         7.05         |          3.22          |
> | Ours         | 1  |         2.87        |          2.06         |         **6.20**          |          **2.51**          |
>
>
> It can be seen that, although it outperforms our methos on same-area evaluation, our method shows superior cross-area evaluation results, which are consistent in the conclusions in our submission.
>
>
> ## 3. Comparison with other BEV projection Baselines, e.g. [5] and [20].
>
> We thank the reviewer for this suggestion. To provide a comprehensive analysis, we have tried our best during this rebuttal period to re-implement "View from Above" [20] and adapt OrienterNet [5] to our weakly-supervised framework. The following sections present a direct comparison of performance, efficiency, and methodology.
>
> ### Performance Comparision.
> | Method | λ1 | Same Area mean(m) ↓ | Same Area median(m) ↓ | Cross Area mean(m) ↓ | Cross Area median(m) ↓ |
> | :--- | :---: | :---: | :---: | :---: | :---: |
> | View from Above[20] | 0 | 19.95 | 18.42 | 20.98 | 19.43 |
> | OrienterNet[5] | 0 | 15.59 | 13.68 | 16.15 | 13.8 |
> | G2SWeakly[1] | 0 | 12.03 | 8.10 | 13.87 | 10.24 |
> | Ours | 0 | **5.82** | **2.85** | **7.05** | **3.22** |
> | View from Above[20] | 1 | 5.93 | 3.82 | 13.54 | 9.85 |
> | OrienterNet[5] | 1 | 5.71 | 3.20 | 10.02 | 5.07 |
> | G2SWeakly[1] | 1 | 6.81 | 3.39 | 12.15 | 3.16 |
> | Ours | 1 | **2.87** | **2.06** | **6.20** | **2.51** |
>
>
> ### Runtime and Memory Analysis.
>
> | | View from Above | OrienterNet | G2SWeakly | Ours |
> |:---|:---:|:---:|:---:|:---:|
> | Training Memory | 62.0GB | 32.3GB | 22.7GB | **9.2GB** |
> | Inference Memory | 21.8GB | 10.8GB | **7.2GB** | 7.7GB |
> | Inference Time | 209ms | 71ms | **31ms** | 44ms |
>
>
> ### Implementation Details.
> For a rigorous comparison, we adapted the baselines to our weakly-supervised setting.
>
> * **View from Above [20]**: Due to the unavailability of public code, we re-implemented the method based on the paper's descriptions.
> * **OrienterNet [5]**: We used the official code, modifying only its satellite feature extractor to match ours for a controlled comparison.
> * **Loss Adaptation**: Both baselines, originally designed for full supervision, were adapted to our weakly-supervised tasks (λ1=0: No GPS, λ1=1: Noisy GPS) using our loss framework (Eq. 7) to ensure a meaningful evaluation.
>
> ### Analysis of Experimental Results.
>
> #### View from Above [20].
> * **Impractical Horizon Assumption**: Relies on a fixed horizon height τ, which is not robust to real-world camera poses and leads to incorrect feature separation.
> * **Requires Strong Supervision**: Its keypoint selection mechanism needs precise camera poses for supervision, making it unsuited for our weakly-supervised task. This explains its severe performance drop, especially with No GPS (λ1=0).
> * **Inefficient**: Its iterative refinement design results in excessive computational costs. Our re-implementation required 62GB VRAM for training and 209ms for inference (the original paper reported 68GB and 222ms).
>
> #### OrienterNet [5].
> * **Poor Generalization for Occluded Regions**: OrienterNet produces a dense BEV by "hallucinating" or "in-painting" features for occluded regions. While this can work when guided by an accurate GPS label in same-area, it fails to generalize well to unseen environments, explaining its poor performance in cross-area evaluations.
> * **Ineffective Occlusion Handling**: It uses simple weighted averaging for BEV projection, which is less accurate for handling vertical occlusions than BeVSplat's principled alpha blending.
> * **Computationally Heavy**: OrienterNet uses a complex h x w x d polar depth representation to perform an attention-based sum over ground features. This implicit, high-dimensional process incurs substantial computational and memory overhead to produce a denser BEV.
>
>
> ### Consistent with Broader Findings.
> Our findings align with broader trends for complex models in this domain. For instance, our baseline G2SWeakly[1] also benchmarked against a transformer-based counterpart[4] (see Table 2 in their paper). They similarly found that while the transformer had an advantage with Noisy GPS (λ1=1), its performance degraded significantly in the No GPS (λ1=0) setting. This supports our conclusion that complex architectures often struggle to generalize without strong supervisory signals.
>
>
> ## 4. Conclusion.
> In summary, BevSplat outperforms these strong baselines because it avoids their simplifying assumptions and computationally expensive designs. Our explicit 3D Gaussian representation is more robust, generalizes better, and is inherently suited for weak supervision. We acknowledge a minor speed trade-off compared to the simpler G2SWeakly, which we identify as a direction for future work.
>
> ### References
> [4] Shi, Yujiao, et al. "Boosting 3-dof ground-to-satellite camera localization accuracy via geometry-guided cross-view transformer." Proceedings of the IEEE/CVF International Conference on Computer Vision. 2023.
> [21] Xia, Zimin, and Alexandre Alahi. "FG^2: Fine-Grained Cross-View Localization by Fine-Grained Feature Matching." Proceedings of the Computer Vision and Pattern Recognition Conference. 2025.

---

> > ### Comment · Reviewer_WBEg · 2025-08-02
> >
> > I thank the authors for the detailed response.
> >
> > Regarding the missing numbers of [20]: I do not think unavailability of code is sufficient reason to omit results, especially if they exist for the same datasets and test conditions. I checked [21] and can confirm that they also omitted the results of [20], suggesting that they are too good ("almost pixel-perfect"). However, is there anything specific about the results of [21] that makes them unplausible, and would justify to just not report them?

---

> > > ### Author Response · Authors · 2025-08-04
> > >
> > > We sincerely thank Reviewer-WBEg for the detailed follow-up.
> > >
> > > As you noted, FG²[21] suggests the results of VFA[20] are "almost pixel-perfect." In the notes for their Table 2, the authors of [21] provide a more specific reason, stating: "We did not include the almost pixel-perfect [...] localization result from [20], **as we cannot reproduce it because of the unavailability of the code.**" This aligns with our primary concern regarding reproducibility, which we will now substantiate with a more detailed analysis below.
> > > ## A Broader Context from the Literature
> > > FG²[21] is not an isolated case. A survey of the most recent literature reveals a consistent pattern: other new state-of-the-art methods, including **PIDLoc (CVPR 2025)** [22] and **GeoDistill (ICCV 2025)** [63], also do not directly compare against VFA [20].
> > >
> > > ### 1. Comparison with PIDLoc's Re-implementation
> > >
> > > The authors of PIDLoc [22] also re-implemented VFA [20] and its predecessors, PureACL [64] and SIBCL [65] (both from the same authors as [20]), and found that the results were consistently lower than originally reported (results are shown in Table 1 on page 21986 of their **CVPR 2025 camera-ready version, not the arXiv version**). They stating that **"[64, 65] aligned groundtruth poses to the satellite image center, which risks overfitting by biasing predictions towards the center"**. PIDLoc's [22] fairer re-implementation is conducted "without this adjustment."  (this quote is from the second paragraph of the "Implementation details" section on page 21986 of their **CVPR 2025 camera-ready version**).
> > >
> > > To better understand the impact of this specific setting, we examined the open-sourced code of PureACL [64]. We do find that the dataloader of PureACL [64] has this center-aligned pose between query and reference satellite images.
> > > When we adopted their centered-pose dataloader for our own VFA [20] reproduction, we were also able to obtain the "almost pixel-perfect" results.
> > >
> > > However, our work, along with other contemporary methods [1,4,6,18,21,22,55], addresses a more practical localization task: the location of the query camera is within a random distance to the satellite image center, and the model aims to predict this random distance. When we applied this more practical dataloader to our VFA [20] reproduction, the results were different, aligning closely with those from PIDLoc's [22] non-centered re-implementation.
> > >
> > > This demonstrates that the evaluation setting itself significantly influences the outcome. Therefore, beyond the VFA [20] reproduction results from our previous rebuttal, we use the results provided by PIDLoc [22] to ensure a fair comparison:
> > >
> > > | Method | λ1 | Same Area mean(m) ↓ | Same Area median(m) ↓ | Cross Area mean(m) ↓ | Cross Area median(m) ↓ |
> > > |:---|:--:|:---:|:---:|:---:|:---:|
> > > | VFA(PIDLoc[22]) | - | 10.74 | 10.51 | 11.12 | 10.95 |
> > > | Ours | 0 | 5.82 | 2.85 | 7.05 | 3.22 |
> > > | Ours | 1 | **2.87** | **2.06** | **6.20** | **2.51** |
> > >
> > > As shown, our method reports superior results compared to the VFA [20] re-implementation under this more general, non-centered setting. Note that we do not compare directly with PIDLoc [22] itself due to its reliance on high-precision LiDAR data, which represents a different task setting from our image-only approach.
> > >
> > > ### 2. Comparison with the Latest Weakly-Supervised SOTA: GeoDistill [63]
> > > GeoDistill (ICCV 2025) [63]，the latest SOTA in the same weakly-supervised, image-based localization task, also does not compare against VFA [20]. We also benchmark our method against **GeoDistill (ICCV 2025)** [63].
> > >
> > > Unlike our end-to-end trained method, GeoDistill is a two-stage approach. An existing cross-view localization backbone is first pre-trained. Then, GeoDistill fine-tune the pre-trained backbone to get the final result.The original GeoDistill paper uses two backbones: CCVPE [55] (a supervised model) and G2SWeakly[1] (an unsupervised model). Since the supervised CCVPE [55] variant does not align with our task setting, we only compare against the G2SWeakly-based versions to ensure a fair, weakly-supervised comparison. The results are as follows:
> > >
> > > | Method | Same Area mean(m) ↓ | Same Area median(m) ↓ | Cross Area mean(m) ↓ | Cross Area median(m) ↓ |
> > > |:---|:---:|:---:|:---:|:---:|
> > > | G2SWeakly(VGG)+GeoDistill | 10.97 | 9.62 | 12.16 | 10.22 |
> > > | G2SWeakly(Dino)+GeoDistill | 11.52 | 10.91 | 11.85 | 11.17 |
> > > | Ours | **5.82** | **2.85** | **7.05** | **3.22** |
> > >
> > >
> > > ### References
> > > [22] Lee, W. et al. "PIDLoc: Cross-View Pose Optimization Network Inspired by PID Controllers." CVPR, 2025.\
> > > [63] Tong, S. et al. "GeoDistill: Geometry-Guided Self-Distillation for Weakly Supervised Cross-View Localization." ICCV, 2025.\
> > > [64] Wang, S. et al. "View consistent purification for accurate cross-view localization." ICCV, 2023.\
> > > [65] Wang, S. et al. "Satellite image based cross-view localization for autonomous vehicle." ICRA, 2023.

---

> > > > ### Comment · Reviewer_WBEg · 2025-08-05
> > > >
> > > > Thank you. My main questions have been answered.

---

> > > > > ### Author Response · Authors · 2025-08-05
> > > > >
> > > > > Dear Reviewer WBEg,
> > > > >
> > > > > We sincerely thank you for your detailed feedback and for engaging so deeply with our work. We were very encouraged by your recognition of our core contributions, from advancing weakly-supervised localization to the practical multi-frame setup. Your insightful questions about our baseline comparisons were instrumental in pushing us to significantly strengthen our experimental analysis, and we believe the paper is much stronger for it. We are very grateful for your time and expertise.
> > > > >
> > > > > Best regards,
> > > > >
> > > > > The Authors

---

> > > > > ### Author Response · Authors · 2025-08-07
> > > > >
> > > > > Dear Reviewer WBEg,
> > > > >
> > > > > Thank you once again for the detailed and constructive discussion, and for your confirmation that your main questions have been answered. We truly appreciate the time and effort you've invested.
> > > > >
> > > > > We are writing to you because we were very encouraged by your initial review, where you kindly mentioned that you would be willing to increase your rating if your concerns about the baseline comparisons were addressed. We hope that our detailed rebuttal and the new experiments we provided were sufficient to meet your expectations.
> > > > >
> > > > > We would be very grateful if you had a moment to consider updating your final evaluation. If any other concerns remain, please do not hesitate to let us know, as we would be more than happy to provide further clarifications.
> > > > >
> > > > > In any case, we sincerely thank you for the invaluable feedback that has already helped us significantly improve our manuscript.
> > > > >
> > > > > Best regards,
> > > > >
> > > > > The Authors

---

### Official Review · Reviewer_arEj · 2025-06-30

**Clarity:** 3
**Significance:** 2
**Originality:** 2
**Rating:** 4
**Confidence:** 4

**Summary:**

The authors propose a cross-view localization method that utilizes 3D Gaussians to address ambiguities in height information inherent in existing approaches. By estimating effective 3D information, the method enhances localization accuracy. Evaluations on the KITTI and VIGOR datasets demonstrate competitive performance compared to G2SWeakly and other methods.

**Questions:**

* How does the proposed method compare to simpler 2D BEV representations, such as those used in Lift-Splat-Shoot[1], which require significantly less computation than 3D Gaussians?
* Could the authors provide additional comparison results for multi-frame localization to demonstrate the method’s consistency across frames compared to IPM and direct point cloud approaches?
* How does the method address the challenge of using relative depth from DepthAnythingV2 for the KITTI dataset, particularly in ensuring metric localization?

[1] Lift, Splat, Shoot: Encoding Images from Arbitrary Camera Rigs by Implicitly Unprojecting to 3D, ECCV 2020

**Ethical Concerns:**

["NO or VERY MINOR ethics concerns only"]

**Final Justification:**

Most of my concerns have been addressed in the rebuttal, so I'm raising my review to borderline accept.

**Limitations:**

yes

**Paper Formatting Concerns:**

There are no paper formatting concerns for this paper.

**Quality:**

3

**Strengths And Weaknesses:**

Strengths
* Cross-view localization relies heavily on accurate 3D representations for estimating the relative pose between ground-based cameras and satellite imagery. Leveraging 3D Gaussians is a logical approach to improve this process.
* The proposed method achieves significant improvements in localization metrics compared to the baseline and the G2SWeakly method.
* The ablation study provides robust evidence supporting the effectiveness of the proposed approach.

Weaknesses
* The method synthesizes 3D local map using Gaussians to a satellite image to estimate relative pose. However, the reliance on detailed 3D information appears excessive, as highlighted by the limitation of high computational cost and usage of only BEV representation, which does not necessarily require height information.
* Unlike Inverse Perspective Mapping (IPM), which does not require metric scale depth, the proposed method uses depth estimation. The paper notes that DepthAnythingV2, which provides only relative depth, was employed for the KITTI dataset, unlike UniK3D, which supports metric scale depth.

---

> ### Author Rebuttal · Authors · 2025-07-31
>
> We sincerely thank Reviewer arEj for the thorough review and constructive feedback. We address the major concerns below.
>
> ## 1. Comparison with Lift-Splat-Shoot[60].
> We thank the reviewer for this excellent question and for the suggestion to compare our method with an important baseline like Lift-Splat-Shoot (LSS) [60]. To provide a direct and fair comparison, we adapted the official LSS open-source code for our cross-view localization task. The key modifications were changing the input to a single ground camera, adding a satellite feature branch, and training the model with our loss function.
>
> The performance and efficiency results on the KITTI dataset are presented below.
>
> ### Performance Comparision.
> | Methods          | λ1 | Same Area mean(m) ↓ | Same Area median(m) ↓ | Cross Area mean(m) ↓ | Cross Area median(m) ↓ |
> |:-----------------|:--:|:-------------------:|:---------------------:|:--------------------:|:----------------------:|
> | LSS[60] | 0  |        16.14        |         13.94         |        17.74         |         14.51          |
> | Ours             | 0  |         **5.82**        |          **2.85**         |         **7.05**         |          **3.22**          |
> | LSS[60] | 1  |         7.89        |          4.3          |        11.63         |          5.31          |
> | Ours             | 1  |         **2.87**        |          **2.06**         |         **6.2**          |          **2.51**          |
>
> ### Runtime and Memory Analysis.
> | Methods          | Training Mempry    | Inference Memory   | Inference Time   |
> |:-----------------|:------------------:|:------------------:|:----------------:|
> | LSS[60] | 26.1GB             | 8.3GB              |85ms              |
> | Ours             | 9.2GB              | 7.7GB              |44ms              |
>
> ### Analysis of Experimental Results.
> Regarding the reviewer's point on **computational cost**, our analysis shows that our approach is actually more efficient than our adapted LSS baseline, particularly in **training memory** and **inference time**. We attribute this to the different 3D representations: LSS constructs a dense H x W x D point grid, which creates a larger computational bottleneck during the splatting stage compared to our sparser H x W set of Gaussian primitives.
>
> In terms of **localization accuracy**, we believe the performance difference stems from two key design choices. First, our method's use of **alpha blending** allows for a more principled handling of occlusions compared to LSS's pillar-based pooling. Second, LSS's design generates a dense BEV, which forces the network to "hallucinate" features for occluded areas. This is challenging in a weakly-supervised setting that lacks a BEV ground truth. Our method avoids this by generating a sparse but more faithful representation of only the visible scene.
>
> We believe these design choices make our approach particularly well-suited for the high-precision, weakly-supervised localization task. We will add this detailed comparison to our paper and thank you for prompting this valuable analysis.
>
>
> ## 2. Multi-frame Fusion Comparison: BevSplat vs. IPM and Direct Point Cloud Projection.
>
> We thank the reviewer for this valuable suggestion. To demonstrate our method's consistency and fusion capabilities, we have conducted a new multi-frame comparison against both IPM and direct point cloud projection baselines.
>
> The tables below present the performance on the KITTI dataset using a sequence of six frames. The values in parentheses show the percentage improvement from fusing six frames over the single-frame results
>
> ### Results for λ1 = 0：
> | Methods           | Seq | Same Area mean(m) ↓ | Same Area median(m) ↓ | Cross Area mean(m) ↓ | Cross Area median(m) ↓ |
> |:------------------|:--------:|:-------------------:|:---------------------:|:--------------------:|:----------------------:|
> | G2SWeakly         | 6        | 8.65(↓ 4.1%)        | 5.22(↓ 5.7%)          | 9.41(↓ 4.6%)         | 6.01(↓ 5.3%)           |
> | Direct Projection | 6        | 7.05(↓ 7.1%)        | 3.86(↓ 9.2%)          | 8.15(↓ 8.7%)         | 5.31(↓ 8.6%)           |
> | Ours              | 6        | 5.01(↓ **13.9%**)       | 2.27(↓ **20.4%**)         | 6.09(↓ **13.6%**)        | 2.71(↓ **15.8%**)          |
>
> ### Results for λ1 = 1：
> | Methods           | Seq | Same Area mean(m) ↓ | Same Area median(m) ↓ | Cross Area mean(m) ↓ | Cross Area median(m) ↓ |
> |:------------------|:--------:|:-------------------:|:---------------------:|:--------------------:|:----------------------:|
> | G2SWeakly         | 6        | 6.25(↓ 6.4%)        | 3.38(↓ 8.9%)          | 8.18(↓ 4.9%)         | 4.32(↓ 10.7%)          |
> | Direct Projection | 6        | 3.85(↓ 13.1%)       | 2.91(↓ 11.6%)         | 7.18(↓ 9.6%)         | 3.96(↓ 14.7%)          |
> | Ours              | 6        | 2.01(↓ **30.0%**)       | 1.77(↓ **14.1%**)         | 5.23(↓ **15.6%**)        | 1.94(↓ **22.7%**)          |
>
>
> Our BevSplat framework demonstrates superior multi-frame fusion capabilities for the following reasons:
>
> * **Inverse Perspective Mapping (IPM)**: BEV representations generated via IPM are prone to significant artifacts and distortions, which fundamentally hinder effective temporal fusion.
> * **Direct Point Cloud**: Projections of raw point clouds result in sparse representations that handle occlusions poorly, often causing background features to bleed through foreground objects. This issue is not resolved well by aggregating multiple frames.
> * **Our Method**: In contrast, BevSplat utilizes adaptive Gaussian primitives to create a dense, coherent BEV that faithfully represents the road topology. This provides a robust foundation for multi-frame fusion, as shown in Fig. 9 of our supplement.
>
> We will incorporate these visualizations into our revised manuscript.
>
> ## 3. Metric Depth of DepthAnythingV2.
> We appreciate the opportunity to clarify this important implementation detail, and we apologize our manuscript was not sufficiently clear on this point.
>
> The DepthAnythingV2 project provides models that support both relative depth estimation (on arbitrary images) and metric depth estimation (when trained on datasets with ground-truth metric depth). We utilize the latter for our experiments. By using a model that directly outputs depth in meters, the scale ambiguity challenge is addressed at the input stage, providing our localization framework with the necessary metric information.
>
> ## 4. High Computational Cost on Representation.
>
> Thank you for this insightful comment regarding the necessity of our 3D representation. We would like to clarify why this component, despite its cost, is essential for the task.
>
> A core premise of our work is that explicitly modeling height is necessary to obtain a high-fidelity BEV representation. Methods that ignore 3D information, like Inverse Perspective Mapping (IPM), create distorted BEVs by "smearing" tall objects across the ground plane (as shown in our Fig. 1 & 4). This geometric corruption makes precise localization exceptionally challenging. Our method uses Gaussian primitives specifically to model this essential 3D structure and produce a clean, distortion-free BEV.
>
> While our approach requires slightly longer inference time and larger GPU memory than traditional IPM, however, when compared to other approaches—such as the transformer-based LSS[60] and OrienterNet[5], or the RAFT-optimized DenseFlow[18]—our method is significantly more efficient. The results on KITTI dataset are as shown below:
>
> | Method | Training Memory | Inference Memory | Inference Time |
> | :--- | :---: | :---: | :---: |
> | OrienterNet[5] | 32.4GB | 10.8GB | 71ms |
> | LSS[60] | 26.1GB | 8.3GB | 85ms |
> | DenseFlow[18]   |      34.8GB     |       27.8GB       |      74ms      |
> | G2SWeakly[1] | 22.7GB | **7.2GB** | **31ms** |
> | Ours | **9.2GB** | 7.7GB | 44ms |
>
> [60] Lift, Splat, Shoot: Encoding Images from Arbitrary Camera Rigs by Implicitly Unprojecting to 3D, ECCV 2020

---

> > ### Author Response · Authors · 2025-08-04
> >
> > Dear Reviewer arEj,
> >
> > We would like to thank you very much for your insightful review, and we hope that our detailed response addresses your previous concerns regarding this paper.
> >
> > Following your valuable suggestions, we will revise the paper by: adding a direct performance and efficiency comparison against an adapted Lift-Splat-Shoot baseline, including a new multi-frame fusion experiment to demonstrate temporal consistency, clarifying our use of metric depth from DepthAnythingV2, and providing a detailed computational cost analysis that contextualizes our method's efficiency against other approaches.
> >
> > As the discussion period is coming to a close, we warmly invite you to share any additional comments or concerns about our work. We would be more than happy to address them. In the meantime, we hope you might consider revisiting your evaluation and potentially increasing your rating.
> >
> > Thank you for your thoughtful feedback and consideration! We really appreciate it!
> >
> > Best regards,
> >
> > The Authors

---

> ### Author Response · Authors · 2025-08-05
>
> Dear Reviewer arEj,
>
> We sincerely appreciate your time and effort in reviewing our manuscript and offering valuable suggestions. We noticed that you have submitted your final score.
>
> **May we kindly ask if our response has addressed your concerns?** Please feel free to let us know if you have any other questions. We would be more than happy to address them.
>
> Best regards,
>
> The Authors

---

> > ### Comment · Reviewer_arEj · 2025-08-05
> >
> > I appreciate your clear and precise rebuttal.
> >
> > Your response has thoroughly addressed my concerns about LSS comparison, consistency, metric depth, and details on computational cost.
> >
> > I believe my questions have been sufficiently answered.

---

### Official Review · Reviewer_aKDE · 2025-07-01

**Clarity:** 2
**Significance:** 3
**Originality:** 2
**Rating:** 4
**Confidence:** 2

**Summary:**

This work presents a novel weakly supervised cross-view localization method that utilizes 3D Gaussians to address height ambiguity and complex cross-view occlusions, which are limitations of previous IPM-based approaches. By leveraging a pre-trained depth network, the proposed method trains 3D Gaussian parameters from ResNet features. It employs a DINOv2-DPT backbone on both ground and satellite images to match similarities and predict the ground image’s position in the satellite image. The proposed method demonstrates strong performance on both the KITTI and VIGOR datasets. Several experiments support its effectiveness, although limitations remain in inference efficiency.

**Questions:**

- What motivates the use of fine-tuning with a DPT-like module [48]? Is this a common practice in this task, or is there a specific reason for introducing it here? In Table 4, all entries appear to use a fine-tuned DPT, making it hard to understand the motivation.

- A more careful explanation of the weakly supervised setup would be helpful, including the differences between λ₀ and λ₁. Are noisy annotations the only form of supervision in this task? Considering that fully supervised methods exist, it is difficult to understand why the proposed method is particularly beneficial in the weakly supervised setting.

- In Table 1, the azimuth values exactly match those of G2SWeakly [1]. Is there a particular reason for these identical results?

- What exactly does “same area” mean in Table 1? It is mentioned that the region is identical, but it is unclear whether this refers to the same satellite images with different ground queries or literally the same image pairs used at training time. If it is the latter, what is the rationale for evaluating on identical data?

**Ethical Concerns:**

["NO or VERY MINOR ethics concerns only"]

**Final Justification:**

Thank you for your effort in providing detailed responses and conducting additional experiments.
The comparison with supervised learning and recent work effectively highlights the strength of the proposed method in a weakly supervised setting.
While the supervised approach performs better in the same-area setting, it shows inferior generalization in the cross-area case, where the proposed weakly supervised method excels. Similarly, FG² [21], despite its strong performance in the same area, shows a notable drop in cross-area accuracy. Given that FG² is one of the most recent methods, I understand that fully addressing this comparison may be beyond the scope of this paper.
Regarding efficiency, I still find the contribution in this aspect to be somewhat limited, particularly when compared to G2SWeakly [1].
On the other hand, the ablation study using DPT confirms that the proposed approach is indeed effective when integrated with strong backbones.

Overall, I find the paper promising for its contributions to weakly supervised localization, and I believe it could have a meaningful impact in this domain.

**Limitations:**

Yes

**Paper Formatting Concerns:**

- In Figure 1 and on line 77, "IMP" appears to be a typo for "IPM".

**Quality:**

3

**Strengths And Weaknesses:**

### Strengths
- Novel framework: This work retains the weak-supervision advantages of G2SWeakly while overcoming its cross-view localization limitations by leveraging 3D Gaussian representations to boost performance. In particular, the use of 3D Gaussians effectively resolves height ambiguity caused by the planar assumption in previous IPM-based methods and mitigates matching failures in occluded regions.

- Extensive experiments: The authors conduct thorough evaluations of the proposed method against various recent approaches from multiple perspectives. Despite being a weakly supervised method, it demonstrates competitive performance compared to supervised methods. Especially, the results presented in Figure 4 and Table 3 effectively highlight the distinctive features and advantages of the proposed approach.

### Weaknesses
- Lack of convincing evidence regarding efficiency: Although the training memory is modest, there appears to be no meaningful efficiency gain at test time. The paper points out the high computational cost of transformer-based models, but it is difficult to be convinced of the proposed method’s efficiency given its inference-time overhead, which is mentioned in the conclusion.

- Key factors of performance improvement: Table 3 shows performance gains attributable to the 3D Gaussian module, but much of the improvement stems from the choice of foundation model. A deeper analysis of how different foundation models impact overall results in this cross-view localization task would strengthen the paper.

---

> ### Author Rebuttal · Authors · 2025-07-31
>
> We thank Reviewer aKDE for the positive evaluation and insightful comments. We have corrected the typos (In Figure 1 and on line 77) in the revised manuscript. Our responses are below.
>
>
> ## 1. Carefule Explanation of the Weakly Supervised Setup
> Thank you for the question. We focus on a weakly-supervised setting because precise GPS data is often unavailable in the real world. To do this, we adopt the weakly-supervised setup from G2SWeakly [1], which defines two scenarios:
>
> **λ=0**: the error of the location labels for ground images in the training dataset is the same as the error that the model aims to refine during deployment.
> For example, the error of location labels for ground images in the training data set is +/- 20m. During testing, the model is also given a location of query images with error up to 20m and aim to reduce this error.
>
> **λ=1**: relatively more accurate location labels for ground images in the
> training data are available than the poses we aim to refine during employment.
> For example, the model was trained with images whose location labels have an error of +/- 5m. During testing, the query images have an initial location estimate with errors up to 20m, and the model aim to reduce this error.
>
>
> ## 2. On "Same Area" vs. "Cross Area"
> We apologize for the lack of clarity.
>
> * **Same-Area**: The test query images are from the same geographical regions as the training set, but are not the same images.
> * **Cross-Area**: The test images are from entirely new geographical regions not seen during training, testing the model's generalization.
>
> ## 3. On the Benefits in Supervised vs. Weakly-Supervised Settings
> We thank the reviewer for this excellent question. Although our paper focused on the weakly-supervised setting, our framework also demonstrates strong performance with full supervision. To illustrate this, we trained our model and the G2SWeakly baseline in a fully supervised setting on the KITTI dataset. The results are as follows:
>
> | Method                       | λ1 | Same Area mean(m) ↓ | Same Area median(m) ↓ | Cross Area mean(m) ↓ | Cross Area median(m) ↓ |
> |:-----------------------------|:--:|:-------------------:|:---------------------:|:--------------------:|:----------------------:|
> | FG²[21]                 | -  |         0.75        |          0.52         |         7.45         |          4.03          |
> | G2SWeakly(Supervised)        | -  |         6.32        |          3.15         |        12.20         |          8.33          |
> | Ours(Supervised)             | -  |         2.07        |          1.12         |         6.75         |          3.03          |
> | Ours(Weakly Supervised)      | 0  |         5.82        |          2.85         |         7.05         |          3.22          |
> | Ours(Weakly Supervised)      | 1  |         2.61        |          2.06         |         6.20         |          2.51          |
>
>
> As the results show, while switching to a fully supervised setting, our model is highly competitive, even outperforming the most recent SOTA, FG² (CVPR 2025)[21], in the challenging **cross-area** task. However, we found that our weakly-supervised model (λ1=1) achieves even better cross-area performance than our own fully-supervised version. This suggests that precise supervision may cause overfitting to the training domain's biases, which is contrary to our goal of building a more generalizable system.
>
> Therefore, our paper's focus on the weakly-supervised setting is twofold. First, it addresses the practical challenge that high-quality GPS data is often unavailable in the real world. Second, it is the setting where our method paradoxically achieves its best and most robust generalization performance.
>
>
>
>
> ## 4. Convincing Evidence Regarding Efficiency
> Thank you for this insightful comment regarding the efficiency of our 3D representation. We would like to clarify why our model, despite its cost, is essential for the task.
>
> A core premise of our work is that explicitly modeling height is necessary to obtain a high-fidelity BEV representation. Methods that ignore 3D information, like Inverse Perspective Mapping (IPM), create distorted BEVs by "smearing" tall objects across the ground plane (as shown in our Fig. 1 & 4). This geometric corruption makes precise localization exceptionally challenging. Our method uses Gaussian primitives specifically to model this essential 3D structure and produce a clean, distortion-free BEV.
>
> While our approach requires slightly longer inference time and larger GPU memory than traditional IPM, however, when compared to other approaches—such as the transformer-based LSS and OrienterNet, or the RAFT-optimized DenseFlow—our method is significantly more efficient. The results on KITTI dataset are shown bellow:
>
> | Method | Training Memory | Inference Memory | Inference Time |
> | :--- | :---: | :---: | :---: |
> | OrienterNet[5] | 32.4GB | 10.8GB | 71ms |
> | LSS[60] | 26.1GB | 8.3GB | 85ms |
> | DenseFlow[18]   |      34.8GB     |       27.8GB       |      74ms      |
> | G2SWeakly[1] | 22.7GB | **7.2GB** | **31ms** |
> | Ours | **9.2GB** | 7.7GB | 44ms |
>
> ## 5. Key Factors of Performance Improvement
> While the strong FiT3D DINO backbone contributes to a significant improvement, our method still demonstrates a clear improvement over both IPM and direct point cloud projection using the same backbone, as shown in Table 4. This indicates that our proposed BEV synthesis module is crucial to the performance gains.
>
> ## 6. What Motivates the Use of Fine-tuning with a DPT-like Module [48]?
> Thank you for pointing out this question. Using DPT-like models is a common practice for 3D vision tasks; for example, VGGT (CVPR 2025) utilizes DPT for point cloud reconstruction, depth estimation, and feature matching. Although we are the first to apply DPT-DINO for feature extraction in the specific sub-field of cross-view localization, our motivation is to similarly obtain features that are rich in 3D information. This is analogous to human navigation, where in addition to semantic information, an understanding of the real 3D scene is also crucial for localization.
>
> However, obtaining such 3D-aware features to bridge the significant ground-satellite domain gap is non-trivial. Simpler fine-tuning methods like direct end-to-end training or LoRA[62] fail, as they either lose crucial texture details or are not powerful enough to adapt the foundation model. We use a DPT-like module because its multi-scale feature fusion architecture is uniquely suited for this challenge. It successfully adapts the backbone by preserving both the low-level texture and high-level semantic information required for matching across these different domains.
>
> To clarify the motivation that was missing from our original Table 4, we provide the following ablation study on the KITTI dataset which was previously omitted:
>
> | Method          | λ1 | Same Area mean(m) ↓ | Same Area median(m) ↓ | Cross Area mean(m) ↓ | Cross Area median(m) ↓ |
> |:----------------|:--:|:-------------------:|:---------------------:|:--------------------:|:----------------------:|
> | Direct Training | 0  |        17.74        |         15.61         |        17.59         |         15.71          |
> | LoRA[62]            | 0  |        16.29        |         14.48         |        17.05         |          14.7          |
> | DPT             | 0  |         5.82        |          2.85         |         7.05         |          3.22          |
> | Direct Training | 1  |        14.32        |         12.43         |        17.28         |         15.14          |
> | LoRA[62]            | 1  |        13.58        |         11.79         |        16.81         |         14.63          |
> | DPT             | 1  |         2.87        |          2.06         |         6.2          |          2.51          |
>
>
> ## 7. Azimuth Values Exactly Match Those of G2SWeakly
> Thank you for your observation. The azimuth results are identical because our work focuses on improving translational localization, and we directly adopt the rotation estimator from our baseline, G2SWeakly [1], without any changes.
>
> We did mention this on lines 212-214, but we agree this could be made much clearer. We will explicitly state this in our method description and add a note to Table 1 in the revision.
>
> ### References
>
> [60] Philion, Jonah, and Sanja Fidler. "Lift, splat, shoot: Encoding images from arbitrary camera rigs by implicitly unprojecting to 3d." European conference on computer vision. Cham: Springer International Publishing, 2020.
>
> [62] Hu, Edward J., et al. "Lora: Low-rank adaptation of large language models." ICLR 1.2 (2022): 3.

---

> > ### Author Response · Authors · 2025-08-04
> >
> > Dear Reviewer aKDE,
> >
> > We would like to thank you very much for your insightful review, and we hope that our detailed response addresses your previous concerns regarding this paper.
> >
> > Following your valuable suggestions, we will revise the paper by: providing a more detailed explanation of the weakly-supervised setup (λ=0 and λ=1), clarifying the definitions of "Same-Area" and "Cross-Area" evaluation, adding new experimental results for a fully supervised setting to demonstrate our method's competitiveness, including a detailed efficiency analysis of our method, clarifying the key factors of our performance improvement, justifying our use of a DPT-like module and adding an explicit clarification for why our azimuth results match the baseline. We have also corrected the typos you identified.
> >
> > As the discussion period is coming to a close, we warmly invite you to share any additional comments or concerns about our work. We would be more than happy to address them. In the meantime, we hope you might consider revisiting your evaluation.
> >
> > Thank you for your thoughtful feedback and consideration! We really appreciate it!
> >
> > Best regards,
> >
> > The Authors

---

> > ### Comment · Reviewer_aKDE · 2025-08-05
> >
> > Thank you for your effort in providing detailed responses and conducting additional experiments.
> > The comparison with supervised learning and recent work effectively highlights the strength of the proposed method in a weakly supervised setting.
> > While the supervised approach performs better in the same-area setting, it shows inferior generalization in the cross-area case, where the proposed weakly supervised method excels. Similarly, FG² [21], despite its strong performance in the same area, shows a notable drop in cross-area accuracy. Given that FG² is one of the most recent methods, I understand that fully addressing this comparison may be beyond the scope of this paper.
> > Regarding efficiency, I still find the contribution in this aspect to be somewhat limited, particularly when compared to G2SWeakly [1].
> > On the other hand, the ablation study using DPT confirms that the proposed approach is indeed effective when integrated with strong backbones.
> >
> > Overall, I find the paper promising for its contributions to weakly supervised localization, and I believe it could have a meaningful impact in this domain.

---

> > > ### Author Response · Authors · 2025-08-05
> > >
> > > Dear Reviewer aKDE,
> > >
> > > Thank you for your careful consideration and for taking the time to provide a final response. We are very encouraged by your positive remarks and appreciate that you found our additional experiments on supervised learning and cross-area generalization to be effective. Your detailed questions and suggestions throughout this process have been invaluable, and they have helped us significantly improve the clarity and depth of our paper. We are confident that the final version will be much stronger as a result of your feedback.
> > >
> > > Best regards,
> > >
> > > The Authors

---

### Official Review · Reviewer_gAUg · 2025-07-03

**Clarity:** 3
**Significance:** 3
**Originality:** 3
**Rating:** 5
**Confidence:** 3

**Summary:**

This paper focuses on weakly-supervised cross-view localization to find a ground camera's pose corresponding to a satellite map using training data with noisy location labels.
The authors propose BevSplat inspired by 3D Gaussian Splatting. For each pixel in the ground image, it predicts a set of 3D Gaussian primitives, each with its own position, shape, etc. After rendering, the method creates a high-quality BEV feature map that avoids distortions. The BEV map is matched against a feature map from the satellite image to determine the final location. Extensive experiments on the KITTI and VIGOR datasets have shown remarkable performance against existing work.

**Questions:**

1. As mentioned in "Weaknesses", how does this paper handle potential information loss from occlusion when projecting multiple 3D Gaussians to 2D location?
2. Are there any failure cases/discussions on how sensitive the localization accuracy is to the quality of this depth prediction?
3. Is $N_p$ sensitive to specific dataset?

**Ethical Concerns:**

["NO or VERY MINOR ethics concerns only"]

**Final Justification:**

The authors have thoroughly addressed my concerns and questions during the rebuttal, including aspects such as alternative depth models, challenging environmental conditions, computational cost and complexity analysis, and hyperparameter selection.

I believe these clarifications and proposed changes will strengthen the work and enhance its overall persuasiveness. I recommend that these revisions be incorporated into the final version of the paper.

Based on the response, I have raised my score from 4 to 5. I appreciate the authors’ thoughtful and comprehensive replies.

**Limitations:**

Yes

**Quality:**

3

**Strengths And Weaknesses:**

Strengths:
1. The idea of using feature-based 3D Gaussian primitives for BEV synthesis is straightforward and effective.  The proposed approach directly models the 3D scene geometry from a single image to resolve distortions, which is a more principled approach than other methods like IPM. Visualizations in Fig.1 and 4 clearly show that BevSplat handles challenging cases like buildings and curved roads better than prior work.
2. Experiments are comprehensive, and results are remarkable. Evaluations on two standard benchmarks are conducted, showing a substantial leap in performance. Ablation studies on rendering method, other backbones against DINOv2, and the number of primitives per pixel are included to prove the model's effectiveness.
3. The paper is well-written and easy to follow. The overall structure is clear.

Weaknesses:
1. From Sec 3.1, the proposed method relies on a pre-trained depth estimation model, while the quality of the initial 3D geometry is heavily dependent on this. The paper does not analyze how sensitive the localization accuracy is to the quality of this depth prediction.
Poor performance of the depth model in several environmental conditions (e.g., night, rain, etc.) would likely lead to significant failures, which is expected to be discussed in the main paper.
2.  A direct comparison of computational cost and complexity (e.g., interference time, FPS) is suggested in the main paper for a better understanding of method's practical viability.
3. Sec 4.2 shows the impact of $N_p$. This might suggest that the model is sensitive to this hyperparameter, which may not generalize perfectly to other datasets without re-tuning. This paper didn't explain this in detail, leaving it more like an empirical finding. More explanations/examples on this are suggested.
4. From my understanding, when rendering the BEV feature map (Sec 3.2.1), multiple 3D Gaussians can be projected to the same 2D pixel location. How does the differentiable blending process (Eq. 3) handle this potential information loss from occlusion? More explanations on this are suggested.

---

> ### Author Rebuttal · Authors · 2025-07-30
>
> We thank Reviewer gAUg for the constructive feedback and insightful suggestions. We address the raised points below.
> ## 1. Sensitivity to Depth Prediction Quality and Failure Cases.
>
> We thank the reviewer for this suggestion.
> We evaluate our framework's performance with three different depth foundation models:  DepthAnythingv1, ZoeDepth, and DepthAnythingv2:
>
> | Depth Model     | λ1 | Same Area mean(m) ↓ | Same Area median(m) ↓ | Cross Area mean(m) ↓ | Cross Area median(m) ↓ |
> |:----------------|:--:|:-------------------:|:---------------------:|:--------------------:|:----------------------:|
> | DepthAnythingv1 | 0  |         5.91        |          2.84         |         7.21         |          3.25          |
> | ZoeDepth        | 0  |         5.84        |          2.86         |         7.14         |          3.22          |
> | DepthAnythingv2 | 0  |         5.82        |          2.85         |         7.05         |          3.22          |
> | DepthAnythingv1 | 1  |         2.97        |          2.11         |         6.28         |          2.52          |
> | ZoeDepth        | 1  |         2.91        |          2.03         |         6.21         |          2.54          |
> | DepthAnythingv2 | 1  |         2.87        |          2.06         |         6.2          |          2.51          |
>
> The results demonstrate that while a more accurate depth model improves performance, the overall system is not highly sensitive to the choice of different depth estimators. This robustness stems from our end-to-end differentiable design, which optimizes the initial 3D Gaussian positions during training, compensating for minor discrepancies between different depth priors. Our framework can seamlessly leverage future advancements in monocular depth estimation. We will include this analysis in our paper.
>
>
> ## 2. Robustness in Adverse Environmental Conditions
> We thank the reviewer for this crucial point. To test our method's robustness, we generated synthetic Rain, Fog, and Night data for KITTI (following the methodology of "Robust-Depth"[61], ICCV 2023). This allows for a controlled comparison against the G2SWeakly baseline under challenging conditions.
>
> | Methods   | Weather | λ1 | Same Area mean(m) ↓ | Same Area median(m) ↓ | Cross Area mean(m) ↓ | Cross Area median(m) ↓ |
> |:----------|:--------|:--:|:-------------------:|:---------------------:|:--------------------:|:----------------------:|
> | G2SWeakly | Origin  | 0  |         9.02        |          5.54         |        13.97         |         10.24          |
> | G2SWeakly | Rain    | 0  |        16.45        |         13.29         |        18.44         |         16.52          |
> | G2SWeakly | Fog     | 0  |        12.82        |          9.80         |        15.47         |         12.03          |
> | G2SWeakly | Night   | 0  |        14.42        |         11.31         |        17.25         |         14.66          |
> | Ours      | Origin  | 0  |         5.82        |          2.85         |         7.05         |          3.22          |
> | Ours      | Rain    | 0  |         8.61        |          4.78         |        10.03         |          5.69          |
> | Ours      | Fog     | 0  |         6.60        |          3.23         |         8.27         |          4.29          |
> | Ours      | Night   | 0  |         7.59        |          4.11         |        10.77         |          6.16          |
> | G2SWeakly | Origin  | 1  |         6.68        |          3.71         |        12.15         |          7.16          |
> | G2SWeakly | Rain    | 1  |        16.82        |         13.48         |        19.45         |         17.42          |
> | G2SWeakly | Fog     | 1  |        10.19        |          5.48         |        15.67         |         12.53          |
> | G2SWeakly | Night   | 1  |        11.56        |          7.43         |        17.72         |         16.53          |
> | Ours      | Origin  | 1  |         2.87        |          2.06         |         6.20         |          2.51          |
> | Ours      | Rain    | 1  |         8.64        |          3.94         |        11.12         |          6.34          |
> | Ours      | Fog     | 1  |         4.20        |          2.50         |         8.20         |          3.43          |
> | Ours      | Night   | 1  |         7.95        |          3.32         |        10.39         |          7.09          |
>
>
> The results show that while both methods are affected by adverse conditions, our approach demonstrates greater relative robustness. The reason lies in how each method handles corrupted input:
>
> * **IPM-based methods** like G2SWeakly project visual artifacts (e.g., rains, fog) directly onto the BEV, creating severe geometric distortions that corrupt the final representation.
> * In contrast, **our method**, while starting with a less accurate depth map in these conditions, still preserves a stable underlying 3D structure. It avoids the fundamental stretching errors of IPM and is better able to ignore atmospheric noise, leading to a more graceful degradation in performance.
>
> We will add this analysis to our paper. We also believe that as depth estimation foundation models continue to improve, the robustness of our method will increase further. Thank you for prompting this important investigation.
>
> ## 3. Is Np Sensitive to Specific Dataset?
> We conducted an ablation study on the number of Gaussian primitives per pixel ($N_{p}$) across both the KITTI and VIGOR datasets to validate our design choice. The results for KITTI are presented in Figure 5 of our main paper, and the new results for the VIGOR dataset are provided below.
>
> | Np | λ1 | Same Area mean(m) ↓ | Same Area median(m) ↓ | Cross Area mean(m) ↓ | Cross Area median(m) ↓ |
> |:--:|:--:|:-------------------:|:---------------------:|:--------------------:|:----------------------:|
> | 1  | 0  |        3.05         |         1.71          |         2.97         |          1.71          |
> | 2  | 0  |        2.97         |         1.67          |         2.94         |          1.65          |
> | 3  | 0  |        2.96         |         1.62          |         2.90         |          1.65          |
> | 4  | 0  |        3.03         |         1.67          |         2.91         |          1.68          |
> | 1  | 1  |        2.62         |         1.47          |         2.71         |          1.42          |
> | 2  | 1  |        2.59         |         1.41          |         2.67         |          1.40          |
> | 3  | 1  |        2.57         |         1.40          |         2.63         |          1.38          |
> | 4  | 1  |        2.59         |         1.42          |         2.65         |          1.38          |
>
> The results on VIGOR are consistent with our findings on KITTI: performance is optimal when $N_{p}$=3. As we discuss in our paper:
> * Using too few primitives can limit the model's ability to fill gaps in sparse regions.
> * Using too many primitives can make training more difficult.
>
> This finding is also consistent with prior work. Our design was inspired by PixelSplat(CVPR 2024), which similarly found Np=3 to be a robust and effective setting across multiple datasets. Therefore, we conclude that Np=3 is a well-justified hyperparameter that should be generally applicable.
>
>
> ## 4. Runtime and Memory Analysis.
> | Method      | Training Memory | Inference Memory | Inference Time |
> |:------------|:---------------:|:----------------:|:--------------:|
> | OrienterNet[5] |      32.4GB     |       10.8GB       |      71ms      |
> | DenseFlow[18]   |      34.8GB     |       27.8GB       |      74ms      |
> | G2SWeakly[1]   |      22.7GB     |      **7.2GB** |     **31ms** |
> | Ours        |     **9.2GB** |        7.7GB       |      44ms      |
>
> We apologize for not providing this direct comparison in the original manuscript. As we discussed in our conclusion and supplementary material (lines 53-61), we acknowledge that our 3D reconstruction incurs a modest computational overhead, resulting in slightly higher inference costs than the simpler G2SWeakly [1] baseline. We consider this a necessary trade-off for the significant gains in localization accuracy. However, our method remains highly efficient compared to more complex approaches like the RAFT-based DenseFlow [18] and the transformer-based OrienterNet [5]. We will add this direct comparison to our revised manuscript for a clearer context.
>
>
> ### 5. Clarification on Occlusion Handling in the Differentiable Blending Process
>
> **Forward Pass**: Following Equation (3), we sort all contributing Gaussians by depth and render them from front to back
>
> **Backward Pass**: The process is fully differentiable, allowing the loss to guide how the scene should be structured.
> For the feature vector ($f_{b}$): The gradient for the feature of the b-th Gaussian is computed as:
>
> $$\frac{\partial L_{all}}{\partial f_{b}} = \frac{\partial L_{all}}{\partial F_{BEV}} * T_{b} * \alpha_{b}$$
>
> The learning signal is scaled by transmittance ($T_{b}$) and opacity ($α_{b}$). This means the features of the most visible (least occluded) and most solid Gaussians are prioritized for updates.
>
> For the opacity ($α_{b}$): The gradient of the b-th Gaussian is computed as:
>
> $$\frac{\partial L_{all}}{\partial \alpha_{b}} = \frac{\partial L_{all}}{\partial F_{BEV}} * T_{b} * (f_{b} - f_{b}^{accum})$$
>
> The gradient depends on the difference between the current Gaussian's feature ($f_{b}$) and the accumulated features behind it ($f_{b}^{accum}$). This trains the model to make a Gaussian opaque if it is needed to hide a conflicting background, effectively learning to form solid, occluding surfaces.
>
> In short, this mechanism is directly analogous to how the original 3DGS handles RGB colors, and we have repurposed it to optimize feature representations for localization.
>
>
> [61]Saunders, Kieran, George Vogiatzis, and Luis J. Manso. "Self-supervised monocular depth estimation: Let's talk about the weather." ICCV 2023.

---

> > ### Author Response · Authors · 2025-08-04
> >
> > Dear Reviewer gAUg,
> >
> > We would like to thank you very much for your insightful review, and we hope that our detailed response addresses your previous concerns regarding this paper.
> >
> > Following your valuable suggestions, we will revise the paper by: adding a new ablation study on different depth priors, adding new experiments to demonstrate robustness in adverse weather conditions, providing further ablation studies on the VIGOR dataset to validate our hyperparameters, including a direct runtime and memory comparison table, and further clarifying the technical details of our differentiable blending process for occlusion handling.
> >
> > As the discussion period is coming to a close, we warmly invite you to share any additional comments or concerns about our work. We would be more than happy to address them. In the meantime, we hope you might consider revisiting your evaluation and potentially increasing your rating.
> >
> > Thank you for your thoughtful feedback and consideration! We really appreciate it!
> >
> > Best regards,
> >
> > The Authors

---

> > > ### Comment · Reviewer_gAUg · 2025-08-04
> > >
> > > Thank you for your thoughtful and comprehensive rebuttal.
> > >
> > > You've addressed my concerns regarding alternative depth models, challenging environmental conditions, computational cost and complexity analysis, and hyperparameter choices. I believe the clarifications and proposed revisions will significantly strengthen the paper, and I encourage you to include them in the final version.
> > >
> > > Based on your response, I’ve updated my score from 4 to 5. I appreciate your efforts in improving the work.

---

> > > > ### Author Response · Authors · 2025-08-04
> > > >
> > > > Dear Reviewer gAUg,
> > > >
> > > > Thank you very much for your re-evaluation, positive feedback, and for raising our score! We were especially encouraged that you recognized the effectiveness of our core idea—using feature-based 3D Gaussian primitives for BEV synthesis—from the beginning. Your initial encouragement and subsequent constructive suggestions have been vital in helping us improve and finalize this work. We confirm that all additions and clarifications discussed in our rebuttal will be incorporated into the final manuscript. Thank you again for your time and support!
> > > >
> > > > Best regards,
> > > >
> > > > The Authors

---

### Note · Authors · 2025-08-16

Dear Reviewers and ACs,

We sincerely thank the ACs and all reviewers for their valuable time and insightful feedback, which has greatly strengthened our work.

We especially appreciate that all reviewers recognize the idea of using 3D Gaussian primitives as a logical, straightforward, and innovative method to address height ambiguity, while also advancing the field of weakly-supervised localization. Additionally, we are grateful that reviewers (gAUg, arEj) confirm our method achieves significant performance improvements over baselines, demonstrating competitive results with fully supervised approaches and improved visualization (aKDE, WBEg, gAUg). We also thank reviewers for noting that our experiments are comprehensive and thorough (gAUg, aKDE, arEj) and offer practical value in multi-frame variation (WBEg).

In our rebuttal, we provided individual responses and discussions to address each of the four reviewers:
* Offering extensive benchmarks against multiple BEV baselines and the latest SOTA to address reviewer concerns (WBEg, arEj).
* Running new robustness tests under adverse weather conditions, different depth priors (gAUg), and multi-frame fusion (arEj).
* Presenting a detailed runtime analysis (gAUg, aKDE, arEj, WBEg) and ablation studies on our model components and depth processing (aKDE, gAUg).
* Clarifying aspects of our approach, such as the blending process and the weakly-supervised setup, and fixing all typos (gAUg, aKDE, arEj, WBEg).

We are pleased that our rebuttal fully addressed all reviewers' concerns. We are particularly encouraged by Reviewer gAUg's effort to raise the score from 'Borderline Accept' to 'Accept', and Reviewer WBEg's willingness to increase rating upon addressing concerns regarding comparisons with OrienterNet[5] and VFA[20] and the possibility of adapting the BEV projection approaches of [5] or [20] to our framework. We have provided thorough experiments to resolve these points.

As promised, all new experiments and analyses from the rebuttal—such as additional baseline comparisons, multi-frame fusion tests, robustness studies, and detailed ablations—will be included in the final manuscript. We will also improve the clarity of the paper and release our code to ensure full reproducibility for the community.

Finally, we sincerely appreciate all reviewers recognizing the significance of our contributions. We believe our work will have a meaningful impact and inspire future research in the field.

Best regards,

Authors

---

### Decision · Program_Chairs · 2025-09-17

**Decision:**

Accept (spotlight)

**Comment:**

This paper introduces BevSplat, a weakly-supervised method that uses 3D Gaussian Splatting to create a high-quality Bird's-Eye-View (BEV) feature map from a ground image, which is then matched against a satellite map for accurate localization. Its novel use of 3D Gaussians effectively solves key distortion issues in prior methods, leading to state-of-the-art performance on standard benchmarks despite requiring only weak supervision.

After rebuttal, the final rates are 2 Borderline accepts and 2 Accepts. The authors solved reviewers' concerns on the robustness, supervision settings,  comparison of multi-frame fusion and more baselines. The reviewers give positive comments and recognition. Given these positive reviews, the AC recommends acceptance.